# Particles of a de Sitter Universe

Gizem Şengör 

Department of Physics, Boğaziçi University, Bebek, Istanbul 34342, Turkey; gizem.sengor@boun.edu.tr

**Abstract:** The de Sitter spacetime is a maximally symmetric spacetime. It is one of the vacuum solutions to Einstein equations with a cosmological constant. It is the solution with a positive cosmological constant and describes a universe undergoing accelerated expansion. Among the possible signs for a cosmological constant, this solution is relevant for primordial and late-time cosmology. In the case of a zero cosmological constant, studies on the representations of its isometry group have led to a broader understanding of particle physics. The isometry group of $d + 1$-dimensional de Sitter is the group $SO(d + 1, 1)$, whose representations are well known. Given this insight, what can we learn about the elementary degrees of freedom in a four dimensional de Sitter universe by exploring how the unitary irreducible representations of $SO(4, 1)$ present themselves in cosmological setups? This article aims to summarize recent advances along this line that benefit towards a broader understanding of quantum field theory and holography at different signs of the cosmological constant. Particular focus is given to the manifestation of $SO(4, 1)$ representations at the late-time boundary of de Sitter. The discussion is concluded by pointing towards future questions at the late-time boundary and the static patch with a focus on the representations.

**Keywords:** de Sitter spacetime; $SO(4, 1)$ representations; wavefunction; two-point functions; late-time boundary



## 1. Introduction

Symmetries give us a way to digest why nature is the way it is. The symmetries that help us reconcile with our understanding of particles are the coordinate transformations that make up the Poincaré group, $ISO(3, 1)$ [1]. This group involves the whole list of coordinate transformations that leave the four dimensional flat spacetime which has a vanishing cosmological constant, invariant. Observations from the primordial universe, as well as the current day universe indicate the presence of a positive cosmological constant in these two epochs, at large distances. Even if there be tensions in the exact numerical value of the cosmological constant, these observations agree on the sign of it. The spacetime with a positive cosmological constant is de Sitter spacetime. The de Sitter spacetime also has a well established set of coordinate transformations that leave it invariant. The symmetries of the de Sitter spacetime make up the de Sitter group, $SO(4, 1)$.

In the case of the vanishing cosmological constant, studying the unitary irreducible representations of the Poincaré group have guided us in terms of how many degrees of freedom to expect for massive and massless particles with a given spin. So far, these predictions have stood well against tests with experiments at particle colliders and have formed our current understanding of quantum field theory on flat spacetime.

In this review, we would like to point out the properties and representations of the de Sitter group, which may help invoke new ways to approach the presence of the positive cosmological constant. One place where the unitary irreducible representations of the de Sitter group can easily be recognized is at the late-time boundary of de Sitter. This is also the slice which resembles where inflationary observables from the primordial universe live. Motivated by these four dimensional observations from particle physics and cosmology, we will work in four spacetime dimensions in this review. We will support our discussion with examples from the late-time boundary of de Sitter involving free scalar fields.

The unitary irreducible representations of the de Sitter group fall under the following categories: *principal series, complementary series, exceptional series, discrete series*. In four spacetime dimensions, focusing on scalars alone provides a relatively easy, yet adequately equipped setup for recognizing various categories of the de Sitter representations. It will only be the category of exceptional series that we do not get to discuss. The examples will be in terms of the *late-time operators* introduced and recognized as unitary irreducible representations of the de Sitter group for general dimensions in [2], and their two-point functions further studied in [3]. Within the review, we will also point out references that deal with non-zero spin and interactions.

The Poincaré and the de Sitter groups are both noncompact groups, which makes them more intricate to study compared to compact groups. Both of these groups are realized in physical situations. Establishments in the representation theory of the de Sitter group in general dimensions mainly rely on the works of Harish-Chandra in mathematics literature [4–9]. Reference [10] is an introductory and pedagogical review on the development of the subject. In our review, we will mainly follow the monograph [11], which focuses on the construction of representations from group elements. Other recent reviews involve [12], which proposes a limit to flat space physics, and [13], which discusses cases with various spin as well as provides an algebraic construction of the representations from the group algebra and makes connections with the constructions in conformal field theory literature. Both of these later references compute characters of the representations and compare the separate cases of unitary irreducible representations in the presence of negative and positive cosmological constants. Another recent review which compares quantum field theory in the cases of zero and positive cosmological constants is that of [14] where group contractions and concepts of being *massive* and *massless* are also considered along other points. Our goal in this review is to draw attention to some recent directions in the discussion of de Sitter representations and their recognition in physical setups in a focused manner. Providing a full list of references on the historical developments of the subject is beyond the aim of this review.

We start our discussion with a summary of de Sitter geometry and its Killing vectors in Section 2. One of the main messages of this section is the lack of global time translation invariance in the case of a positive cosmological constant. We leave the technical details behind this message to Appendix A.1. In Appendix A.2, we discuss the ambient space formalism and point towards relevant references. For the purpose of our discussion, this formalism allows for a convenient way to compare the case of the positive cosmological constant with that of the zero cosmological constant.

In Section 3, we give a brief summary on the representation theory of the de Sitter group. These representations fall under three different categories for $SO(4,1)$ relevant to four dimensional de Sitter. Different categories are designated by the range of the so called *scaling weight*, $c$. We introduce this label, the scaling weight, in the main body of the discussion. Each category also involves a label related to spin. We focus on the normalizability properties of the representations and the properties of the intertwining operators that play an important role on this matter to motivate the different categories of representations that exist.

In Section 4, we introduce the *late-time operators* into our discussion which exhibit all the properties we discussed of the representations in Section 3. For instance, these late-time operators are normalized with respect to the inner products of Section 3. We discuss the late-time operators themselves as well as their two point functions. We denote a general late-time operator by $\mathcal{O}$. In four spacetime dimensions, these operators have dimensions of the form $\Delta = \frac{3}{2} \pm \mu_p$, where $\mu_p$ can be either real or purely imaginary. When it becomes important to distinguish the dimensions, we denote the operator with the lower dimension $\Delta = \frac{3}{2} - \mu_p$ by $\alpha$ and the operator with the higher dimension $\Delta = \frac{3}{2} + \mu_p$ by $\beta$. We use a superscript $L$, $H$ and $M$ to denote if the operator corresponds to a *light* or *heavy* field whose mass, respectively, belongs to the ranges $m < \frac{3}{2}H$ and $m > \frac{3}{2}H$. We reserve the superscript $M$ to denote the field with mass zero. We refer to this field as the massless field;

however, the concept of *masslessness* on de Sitter is far more involved beyond the case of zero mass scalar. The review [14] involves a better summary on the concept of masslessness.

The late-time solution for the mode functions of the massless scalar is studied within the light fields, yet from among the representation categories, it belongs to the discrete series representations while light fields belong to complementary series representations in general. We discuss the scalar field with zero mass in more detail in Section 4.2. We indicate the normalized late-time operator by a subscript $N$. Therefore, throughout the text there are the late-time operators $\alpha^L$, $\beta^L$, $\alpha^M$, $\beta^M$, $\alpha^H$, $\beta^H$ prior to being normalized and the normalized late-time operators $\alpha^L_N$, $\beta^L_N$, $\alpha^M_N$, $\beta^M_N$, $\alpha^H_N$, $\beta^H_N$.

In Section 5, we point out new insights on holographic properties of de Sitter by focusing on the wavefunction picture. Our discussion relies on being able to track the contribution of late-time operators to late-time two point functions of fields and conjugate momenta in cannonical quantization and in wavefunction picture.

We conclude in Section 6 by pointing towards recent advances on spectral decomposition, cases of fermions and gauge fields, the role of characters on the calculations of de Sitter horizon entropy and discussions on the unitarity of interactions on de Sitter. Each of these venues rely on representations in the presence of a positive cosmological constant.

Our convention for the metric signature is mostly plus and we mainly focus on the global and Poincaré patches of de Sitter.

## 2. The de Sitter Geometry and Symmetries

In four dimensions, the de Sitter metric in global coordinates is

$$ds^2_{gl} = -dT^2 + \frac{1}{H^2}cosh^2(HT)d\Omega_3^2 \tag{1}$$

where

$$d\Omega_3^2 = d\theta_1^2 + sin^2\theta_1\left[d\theta_2^2 + sin^2\theta_2 d\theta_3^2\right], \tag{2}$$

is the metric on three sphere, and the coordinates lie in the following ranges $-\infty < T < \infty$, $0 \le \theta_1 < \pi$, $0 \le \theta_2 < \pi$, $0 \le \theta_3 < 2\pi$ [15]. The spatial sections of this geometry are spheres that grow with time in the range $0 \le T < \infty$, due to the behaviour of the scale factor $cosh(HT)$. These coordinates cover more than what is accessible to a single observer living in de Sitter, or to an observer who has access to a primordial de Sitter phase. We will talk about coordinates for these observers, especially the later one, below in due time. One of the merits of the global coordinates, even though they are not the coordinates of a physical observer, is that they carry information about the global properties of the de Sitter spacetime. One such property we want to emphasize is that translations in global time $T$ is not an isometry of de Sitter. That is, one cannot find a global timelike Killing vector $\xi^{T-tr} = \xi^\mu \partial_\mu = \partial_T$ as a solution to the Killing equations, as explicitly demonstrated in Appendix A.1.

Considering the solutions to Einstein equations with a cosmological constant ($\Lambda$), in the case of $\Lambda_M = 0$, we have the Minkowski spacetime. For vanishing cosmological constant, the coordinate transformation $t \to t + a$ with $a = constant$ leaves the Minkowkski metric

$$ds^2_M = -dt^2 + dx^2 + dy^2 + dz^2, \tag{3}$$

invariant. Time translation is an element of the Poincaré group and the associated Killing vector is an element of its algebra. In the case of $\Lambda_{AdS} < 0$, we have the Anti de Sitter space. For negative cosmological constant, time translation $\tau \to \tau + \mathcal{A}$, is a symmetry of the Anti de Sitter space with metric, which in global coordinates can be written as follows

$$ds^2_A = L^2_{AdS}\left[-(r^2 + 1)d\tau^2 + \frac{dr^2}{r^2 + 1} + r^2 d\Omega_2^2\right], \tag{4}$$

where $d\Omega_2^2$ is the metric on unit two sphere. In the case of de Sitter with $\Lambda_{dS} > 0$, the coordinate transformation $T \to T + A$ with $A$ constant, does not leave the de Sitter metric (1) invariant. Among the solutions of Einstein Equations with a cosmological constant, the positive cosmological constant case seems to be a bit of an outcast for not accommodating time translation invariance. This implies that different rules and notions are at play for quantum fields on geometries with a cosmological constant depending on the sign of the constant. Special care must be given to the case of the positive cosmological constant. What these rules are and especially how they differ in the case of a positive cosmological constant, is an active area of investigation. To name a few, the presence of time translation invariance implies conservation of energy. This sets a lower bound on the possible masses of fields on AdS [16,17]. The presence of time translation invariance also allows for asymptotic states to be defined, which sets the S-matrix scattering programme on Minkowski [18].

The de Sitter and Anti de Sitter spacetimes can be embedded in a flat spacetime of one higher dimension. The differences in the nature of these spacetimes at different signs of the cosmological constant also get highlighted when embedding the non-zero cosmological constant ones into a higher dimensional flat spacetime. This embedding is achieved by the *ambient space formalism*, also referred to as the *embedding space formalism*. Embedding the positive cosmological constant case, that is de Sitter spacetime, requires adding an extra space-like dimension while the negative cosmological constant case, that is Anti de Sitter, requires adding an extra time-like dimension. We give a summary on the details of the embedding space formalism in Appendix A.2 where we also list more detailed references and refer to the appendix whenever we make use of a fact from the embedding space formalism in our discussion. In short, a positive cosmological constant can be embedded into one higher dimensional Minkowski spacetime which is a zero cosmological constant solution, while the negative cosmological constant solution gets embedded into a flat spacetime with two time-like dimensions. While the presence of a global time-like Killing vector is a shared feature between the zero and the negative cosmological constant solution, the embedding space formalism suggests there may be other features shared between Minkowski and de Sitter among the three signs of the cosmological constant. A clue in this direction comes from the fact that the *principal series representations* make an appearance in de Sitter quantum field theory and holography as we will discuss more thoroughly below, as well as in flat space holography [19,20] as part of the unitary irreducible representations in both problems. We will explain the properties of principal series representations in detail and give exemplary cases of their realization in our main discussion. An example that hosts principal series representations in the case of a negative cosmological constant on the other hand is that of [21]. This example involves additional structure, such as having $U(1)$ charge, and is explored for the case of two dimensions which is a special case in terms of the number of dimensions where the two signs of the cosmological constant share the same symmetries.

There is a link between the metric on global de Sitter and the metric on the sphere. In four dimensions, from the metric on the sphere

$$d\Omega_4^2 = d\theta_4^2 + \sin^2\theta_4 \left[ d\theta_1^2 + \sin^2\theta_1 \left[ d\theta_2^2 + \sin^2\theta_2 d\theta_3^2 \right] \right],\tag{5}$$

one can obtain the global metric on de Sitter

$$ds_{gl}^2 = -dT^2 + \frac{1}{H^2}cosh^2(HT)\left[ d\theta_1^2 + sin^2\theta_1 \left[ d\theta_2^2 + sin^2\theta_2 d\theta_3^2 \right] \right]\tag{6}$$

by analytically continuing one of the angular directions into the time direction as $\theta_4 = \frac{\pi}{2} - iT$. What such an analytical continuation implies for unitarity of the representations between the two geometries is better tracked in the embedding space formalism. There is a direct link between the generators and the embedding space coordinates as we highlight in Appendix A.2. Due to this fact the effects of the analytical continuation are transferred to an analytical continuation at the level of the generators. This fact makes it difficult to quickly

recognize what will be a unitary representation on one geometry based on knowledge of what is unitary on the other. References [22–24] are examples that explore this line of investigation further with explicit examples from the case of two dimensions. A complementary approach is that of reference [25] which explicitly discusses which one of the unitary representations on the sphere can be analytically continued into unitary irreducible representations on de Sitter in the case of bosons. In identifying which representations can be analytically continued, special care is given to check the normalization of the de Sitter representations obtained from symmetric tensor spherical harmonics which set up the unitary representations on the sphere. This method is discussed for dimensions higher than two. Reference [26] extends this disscussion to the case of fermions starting with Dirac spinors. One can also relate the cases of positive and negative cosmological constant by analytical continuation as discussed in [27].

　　With all this insight, let us summarize the symmetries that are present in the case of a positive cosmological constant.

　　The isometry group of de Sitter spacetime is a group that has long been studied in mathematics literature, starting with the works of Harish-Chandra [4,6–9]. For de Sitter in general $d + 1$ dimensions, this is the group $SO(d + 1, 1)$. This group also happens to be the conformal group of Eudlidean space in $d$ dimensions. Therefore it is a group of interest both in cosmology and in Euclidean conformal field theory. This is a noncompact group with real parameters. We refer the reader to [22] for a nice comparison of the compact rotation group $SO(3)$ and the group $SO(2, 1)$ which addresses the case of two spacetime dimensions. Here, we will focus on the case of four dimensions relevant to cosmological observations, which brings us to the group $SO(4, 1)$. Most of the results we will discuss can be extended to general dimensions, except for the case of discrete series, as we will highlight where necessary. A self-contained review in general dimensions is the monograph [11], where the representations are built from finite group elements. More recent summaries include [2,13] and reference [28] which focuses on fermions and gauge fields in the discussion.

　　A group $G$ is made up of group elements and an operation defined among them such that, among the group elements, there exists an identity element, the operation is closed and associative and each group element has an inverse under this operation. The group elements correspond to transformations that leave a system of equations of interest invariant. In our case of interest, the group elements are coordinate transformations that leave the metric invariant. Apart from group elements, continuous groups, such as $SO(d + 1, 1)$, have generators $\mathcal{L}^{(B)}$ that satisfy a specific algebra. In this case, the group elements $g \in G$ are parametrized by continuous parameters and are related to the generators of the group algebra via exponential maps. The differentiation of the group element with respect to this continuous parameter gives the generator. One can build further operators by considering combinations of generators in the group algebra. Among such combinations there will be operators which commute with all the generators of the algebra. Such operators are called the Casimir operators. They can be built at different orders and play a special role in that the eigenvalues of the quadratic Casimir label the unitary irreducible representations that correspond to physical states. In essence, the problem of finding possible particles corresponds to the problem of figuring out the eigenvalue spectrum of the Casimir operator. Before giving examples to the particles on a de Sitter universe, in this section we will discuss how they are labelled and categorized.

　　The Killing vectors $\xi^{(B)}$, provide a realization of the group generators $\mathcal{L}^{(B)}$. We use the upper case Latin letter $(B)$ to label the Killing vector and the generator, below this label will stand for dilatations, rotations, etc. What defines the generators is the algebra that they satisfy. The Killing vectors on the other hand are differential operators. The commutations of Killing vectors acting on functions matches the group algebra. This is one way to confirm that the generators can be represented by the Killing vectors. Another way is to consider embedding de Sitter spacetime into flat spacetime of one higher dimension where group generators are easier to recognize. This is called the *embedding space* or *ambient space formalism* and we demonstrate this method in Appendix A.2. While we are more

accustomed to Hermitian operators in Physics, in Mathematics literature, it is more common to use antiHermitian generators, and because this makes the reality of the eigenvalue of the Casimir more explicit, we will also make use of antiHermitian generators which we will denote by $\mathcal{L}^{(B)}$ in which case

$$\left[\mathcal{L}^{(B)}, \phi\right] = -\xi^{(B)}\phi, \tag{7}$$

where the anti Hermitian generator $\mathcal{L}^{(B)}$ gets identified with the Killing vector $-\xi^{(B)}$ [13]. The Hermitian generator can be obtained via $\mathcal{L}_H^{(B)} = i\mathcal{L}^{(B)}$

For fields with spin, one needs to include a spin related term in the compact subgroup generators [28]. We will introduce the compact subgroups shortly below.

In general, a group $G$ will be composed of subgroups whose properties determine the properties of the whole group. Certain subgroups play a distinguished role, especially when talking about unitary irreducible representations. The subgroups of $SO(d + 1, 1)$ are *dilatation*, *special conformal transformations*, *spatial rotations*, *spatial translations* and the *maximally compact subgroup*. Each category of a subgroup corresponds to a category of a Killing vector or a linear combination of them. In this list, spatial rotations and the maximally compact subgroup are the only compact groups. Now let us list the Killing vectors in conformal planar patch coordinates and discuss these subgroups in the case of $SO(4, 1)$ one by one.

*The Dilatation subgroup:* This is the one parameter, noncompact subgroup $SO(1, 1)$ which we will also denote by $A$. The dilatation generator corresponds to the dilatation Killing vector. With unit parameter, this Killing vector is

$$\xi^{(A)} = \eta\partial_\eta + x^j\partial_j. \tag{8}$$

Under a dilatation with parameter $\lambda$ coordinates transform as $(\eta, \vec{x}) \rightarrow (\lambda\eta, \lambda\vec{x})$, and a unitary irreducible representation of the dilatation subgroup transforms as

$$\mathcal{O}(\lambda\vec{x}) = \lambda^{-\Delta}\mathcal{O}(\vec{x}). \tag{9}$$

The exponent $\Delta$ is called the scaling dimension of the representation $\mathcal{O}$. While de Sitter has both temporal and spatial coordinates, the representation $\mathcal{O}$ depends only on spatial coordinates which fits in well with it being recognizable at the late-time boundary, which is a purely spacelike surface.

In the case of $SO(d + 1, 1)$, the scaling dimension has a fixed format. It takes on the form where the number of spatial dimensions enter in as

$$\Delta = \frac{d}{2} + c \tag{10}$$

The component $\frac{d}{2}$ corresponds to the half sum of the restricted positive roots while $c$, called the *scaling weight* [11], involves information about the mass of the field. In general dimensions

$$c = \sqrt{\frac{d^2}{4} - \frac{m^2}{H^2}}, \tag{11}$$

and depending on the mass of the field with respect to both the Hubble parameter and the number of spatial dimensions, the scaling weight can be either purely real or purely imaginary. Both these categories correspond to unitary representations. In what follows, we will denote the scaling weight $c$ as $\pm\nu, \pm i\rho$ where $\nu$ and $\rho$ are positive, when giving examples to different categories. The scaling dimension being allowed to be complex

sets $SO(4,1)$ apart from $SO(3,2)$, even though both groups host dilatations. For $SO(4,1)$ we have

$$\Delta = \frac{3}{2} + c. \tag{12}$$

*The spatial rotation subgroup:* This is the group $M = SO(d)$, in the case of four spacetime dimensions $SO(3)$. It is a compact subgroup with real parameters and antisymmetric generators which for unit parameter along the axis $(\hat{i} \times \hat{j})$ correspond to the Killing vectors

$$\xi^{(M-\hat{i}\times\hat{j})} = x_i\partial_j - x_j\partial_i \tag{13}$$

When working with nontrivial spin, one adds a spin component to the $SO(d)$ and $SO(d+1)$ generators [28].

*Spatial translation subgroup:* We will denote this subgroup by $\tilde{N}$. For unit parameter along the $i^{th}$ direction its generators correspond to the Killing vectors

$$\xi^{(\tilde{N}-i)} = \partial_i. \tag{14}$$

This is another one parameter noncompact subgroup like the dilatation subgroup, and again it is an abelian subgroup.

*Special Conformal Transformations subgroup:* This is a three parameter noncompact subgroup, which we will also denote by $N$. Its generators for unit parameter along the $i$th direction correspond to the Killing vectors

$$\xi^{(N-i)} = (x^2 - \eta^2)\partial_i - 2x_i(\eta\partial_\eta + x^j\partial_j). \tag{15}$$

*Maximally Compact subgroup:* This is the group $K = SO(d+1)$ in a general number of dimensions. In the case of four spacetime dimensions, it is the group $SO(4)$. This group is made up of all of the $SO(3)$ rotation subgroup generators. Additionally, it involves generators which we explicitly list in appendix A.2, that come as a specific combination of translation and special conformal transformation generators. In terms of the Killing vectors the generators of the maximally compact subgroup correspond to

$$\xi^{(M-\hat{i}\times\hat{j})} \text{ and } \frac{1}{2}\left(\xi^{(\tilde{N}-i)} - \xi^{(N-i)}\right). \tag{16}$$

As outlined in more detail in Appendix A.2, the eigenvalue of the quadratic Casimir for $SO(4,1)$ is [11]

$$\mathcal{C} = l(l+1) + c^2 - \frac{9}{4}. \tag{17}$$

Notice that this depends on the spin $l$ and the scaling weight $c$. Hence, the unitary irreducible representations are labeled by spin and scaling weight. This is collectively denoted as $\chi = \{l, c\}$. The value of the scaling weight being purely imaginary, discrete or real within a finite spin-dependent range puts the representation in the category of *principal, discrete and complementary series*. In general, the range of the scaling weight for the complementary series has overlaps with the discrete series range. The unitary representations at these overlaps are referred to as the *exceptional series*. These representations are reducible down to unitary irreducible discrete series representations [11], and for four spacetime dimensions the exceptional series simply corresponds to the discrete series [13]. The eigenvalue of the Casimir gives insight on what category a field of interest belongs to. However, to emphasize the unitarity properties of the representation it hosts, it helps to check normalizability properties and this is what we mainly discuss next.

### 3. Unitarity of the Representations

One example to identify the unitary irreducible representations of $SO(4,1)$ is in the late-time limit of field solutions in the free field theory, that satisfy Bunch–Davies initial conditions. These unitary irreducible representations, let us denote an arbitrary one of them by $\mathcal{O}$, can be written in both the position or the momentum space. The two are related by the usual Fourier transformation

$$\mathcal{O}(\vec{x}) = \int \frac{d^3k}{(2\pi)^3} \mathcal{O}(\vec{k}) e^{i\vec{k}\cdot\vec{x}}. \tag{18}$$

The scaling dimension $\Delta = \frac{3}{2} + c$ is read off from the position space version.

The unitarity of these representations implies having a well-defined finite inner product with which they can be normalized. In order to better appreciate the merits of this inner product, it will be more instructive to understand how these representations are built. There are two ways to build the representations. Either one can construct them as states on which the generators act on or build them from finite group elements. In both ways, unitarity implies having normalizable representations. In the construction from the generators, the unitarity of the states are encoded in their normalization such that different categories have different normalizations. Reference [22] explicitly shows what is the normalization for each category of principal, complementary and discrete series representations in the case of two dimensions; and explains why the normalization for $SO(2,1)$ states is different from the normalization of $SO(3)$ states. This algebraic approach has also been used in recent references to recognize principal series states on $dS_2$ with a holographic counterpart in [29] and on $AdS_2$ from charged scalar fields in [21]. Here, we will focus on the construction from finite group elements following [2,11] where the representations are constructed as maps from group elements to vector spaces. These maps act as homomorphisms on the elements of function spaces built with specific covariance properties as we will summarize below. Both methods have also been summarized in [13].

Among the subgroups of $SO(4,1)$, the combination of special conformal transformations, dilatations and rotations make up the stability subgroup, referred to as the *parabolic subgroup*, $P = NAM$. This discussion also applies to general dimensions, as summarized in [2]. For the group $SO(4,1)$, the parabolic subgroup does the same job that the Little group does in studying the representations of the Poincaré group in the case of a zero cosmological constant. That is, the elementary representations of $SO(4,1)$ are built up from the parabolic subgroup. In essence their behaviour are determined by rotations, dilatations and special conformal transformations. This works into the covariance properties of the function spaces on group elements.

In general, a representation $\Pi$, is a map from group elements $g \in G$, to a vector space $V$ with elements $v \in V$, including the automorphism of this vector space,

$$\Pi : G \rightarrow aut(V) \tag{19}$$

such that this map is a *homomorphism*. That is, for elements $g, g' \in G$ and $v \in V$, the structure is preserved as follows

$$\Pi_g \Pi_{g'} v = \Pi_{gg'} v. \tag{20}$$

Representations induced by the parabolic subgroup are maps from group elements to automorphisms of function spaces where the elements of the function space are infinitely differentiable functions whose domain consists of group elements and whose range is the finite dimensional Hilbert space where the unitary representations of the rotation subgroup are realized. Denoting the rotation group elements by $m \in M = SO(3)$ and labeling the unitary irreducible representations of the rotation subgroup by $l$, we will denote the unitary irreducible representations of the rotation subgroup by $D^l(m)$ and the Hilbert space where

these representations are realized by $\mathcal{V}^l$. With this notation we have functions on group elements $\mathfrak{f}(g)$, such that

$$\mathfrak{f} : G \to \mathcal{V}^l. \tag{21}$$

Moreover, these functions satisfy certain covariance conditions, each of which defines a certain function space. Apart from functions over group elements, function spaces with functions whose domain is the maximally compact subgroup $K = SO(4)$, also enter the discussion. We will denote the function spaces by $C_\chi$ when the domain is the elements of the full group $g \in G = SO(4,1)$ and by $C(K, \mathcal{V}^l)$ when the domain is over the maximally compact subgroup $q \in K = SO(4)$.

The representations induced by the parabolic subgroup are maps $\mathcal{I}_\chi$ from the group to the function space such that they are homomorphisms

$$\mathcal{I}^\chi : G \to C_\chi \text{ such that } (\mathcal{I}_g\mathfrak{f})(g') = \mathfrak{f}\left(g^{-1}g'\right). \tag{22}$$

The structure of the argument $g^{-1}g'$, where $g^{-1}$ denotes the inverse elements, guarantees that the structure is preserved, as in (20).

The covariance conditions determine what happens to functions in the function space under certain transformations. These transformations are carried out by considering the combination of a group element from the domain of the function with elements from subgroups. With the properties of these function spaces written in the format

$$\mathcal{C} = \{\mathfrak{f} : \text{Domain} \to \text{Range}; \text{ covariance condition }\}, \tag{23}$$

the two function spaces of interest are

$$C_\chi = \left\{\mathfrak{f} : G \to \mathcal{V}^l; \ \mathfrak{f}(gnam) = |a|^{\frac{3}{2}+c} D^l(m)^{-1}\mathfrak{f}(g)\right\}, \tag{24}$$

$$C(K, \mathcal{V}^l) = \left\{\mathfrak{f} : K \to \mathcal{V}^l; \ \mathfrak{f}(qna) = |a|^{\frac{3}{2}+c}\mathfrak{f}(q), \ \mathfrak{f}(qm) = D^l(m)^{-1}\mathfrak{f}(q)\right\}. \tag{25}$$

A group element $g \in G$ can be decomposed in terms of elements of subgroups. There are a couple of ways to perform such decompositions, each of which involves different subgroups. Denoting the elements of the subgroups as $m \in M$, $a \in A$, $n \in N$, $\tilde{n} \in \tilde{N}$, $q \in K$, one such decomposition is the *Iwasawa decomposition*, $g = qna$, which highlights contributions from the maximally compact subgroup ($K$). Another one is the *Bruhat decomposition*, $g = \tilde{n}nam$ which highlights contributions from translations ($\tilde{N}$) and rotations ($M$). Due to these decompositions, there is a unique correspondence between the group elements $g \in G$ and the elements of the maximally compact subgroup $q \in K$, which further makes it possible to identify the functionspace $C_\chi$ with the function space $C(K, \mathcal{V}^l)$. Notice that the covariance condition $\mathfrak{f}(qna) = |a|^{\frac{d}{2}+c}\mathfrak{f}(q)$ is the covariance condition of $C_\chi$ for the specific case where $g$ is set to overlap with the maximally compact subgroup element $q$ and $m$ is set to identity. Similarly the covariance condition $\mathfrak{f}(qm) = D^l(m)^{-1}\mathfrak{f}(q)$ is the covariance condition of $C_\chi$ with $g$ again chosen to be $q$ and $na$ set to be the unit element.

While functions over group elements might sound abstract, when in practice we work with spacetime coordinates, there is a connection between the two. This is established via the spatial translations. Considering position as an element of $\mathbb{R}^3$, we will call this the $\vec{x}$−space with $\vec{x} \in \mathbb{R}^3$, there is a unique connection between elements of $\vec{x}$−space and elements of the subgroup spatial translations $\tilde{n} \in \tilde{N}$. A specific element over $\vec{x}$−space corresponds to a specific element $\tilde{n}_{\vec{x}}$ of translation subgroup, such that the functions $f(\vec{x})$ and $\mathfrak{f}(\tilde{n})$ match each other

$$\vec{x} \leftrightarrow \tilde{n}_{\vec{x}} : \ f(\vec{x}) = \mathfrak{f}(\tilde{n}_{\vec{x}}). \tag{26}$$

To complete the whole relation between group elements and $\vec{x}-$space elements, there is also a unique correspondence between a group element $g \in G$ and an $\vec{x}-$space element $\vec{x}_g \in \mathbb{R}^3$ such that

$$g \leftrightarrow \vec{x}_g : \quad g^{-1}\tilde{n}_{\vec{x}} = \tilde{n}_{\vec{x}}n^{-1}a^{-1}m^{-1}. \tag{27}$$

Based on these correspondences and the definition of the representation (22), in $\vec{x}-$space realization, the representations $T^\chi : G \to \mathbb{R}^3$, are homomorphisms that act on functions $f \in \mathcal{C}_\chi$ as

$$\left(T^\chi_g f\right)(\vec{x}) = |a|^{-\frac{3}{2}-c}D^l(m)f(\vec{x}_g); \tag{28}$$

this relation is explicitly worked out in [2]. To summarize so far, we have functions over group elements and functions over position space where the two are related by (26). Moreover, there is a unique correspondence between group elements and elements of the maximally compact subgroup as well as a unique correspondence between group elements and elements of $\vec{x}$-space.

*The Inner Product*

The function space $\mathcal{C}_\chi$ can be completed into a Hilbert space $\mathcal{H}_\chi$ by further equipping it with a well-defined inner product, denoted by $(,)$, which inherits the properties of the rotation invariant inner product, denoted by $\langle \, | \, \rangle$. In position space, for functions $f_1, f_2 \in \mathcal{C}_\chi$ this inner product is

$$(f_1, f_2) = \int d^3x \langle f_1(\vec{x}) | f_2(\vec{x}) \rangle. \tag{29}$$

The definition of a *unitary representation* is that it preserves this inner product. Meaning, given (29), the unitary representation should satisfy

$$\left(T^\chi_g f_1, T^\chi_g f_2\right) = \int d^3x_g \langle f_1(\vec{x}_g) | f_2(\vec{x}_g) \rangle. \tag{30}$$

Working out the left hand side of this equation gives [2]

$$\int d^3x_g \langle f_1(x_g), f_2(x_g) \rangle |a|^{-(c^*+c)} \overset{!}{=} \int d^3x_g \langle f_1(\vec{x}_g) | f_2(\vec{x}_g) \rangle \tag{31}$$

where $c^*$ is the complex conjugate of $c$. Here, due to (27) the volume element transforms as $|a|^{-3}d^3x = d^3x_g$.

Notice that if the scaling weight is purely imaginary, $c = i\rho, \rho \in \mathbb{R}$, then this relation is automatically satisfied. The case of a purely imaginary scaling weight captures the category of the *unitary principal series representations*. Looking at (11) scalar fields of mass $\frac{3H}{2} < m$ on $dS_4$ belong to the principal series category. We will refer to these fields as *heavy*.

The fact that complex scaling dimensions can host unitary irreducible representations is counterintuitive if intuition is based on the case of a negative cosmological constant. This difference between the two signs of the cosmological constant is due to the fact that the positive cosmological constant case lacks global time translation invariance while it is present in the negative cosmological constant case, bringing along further conditions.

The inner product (29), defined in $\vec{x}-$space, can also be written in momentum space via Fourier transformation

$$(f_1, f_2) = \int \frac{d^3k}{(2\pi)^3} \langle f_1(\vec{k}) | f_2(\vec{k}) \rangle. \tag{32}$$

In the momentum space, the scaling weight determines the overall momentum dependence [2]. The late-time operator $\mathcal{O}(\vec{k})$ corresponds to the unitary irreducible representation

$(T\chi_g f)(\vec{k})$. These representations are built from annihilation and creation operators. Overall they have the following form

$$\mathcal{O}(\vec{k}) = f(a_{\vec{k}}, a_{\vec{k}}^\dagger)k^c \tag{33}$$

where acting on the vacuum state $|0\rangle$,

$$a_{\vec{k}}|0\rangle = 0. \tag{34}$$

The function $f(a_{\vec{k}}, a_{\vec{k}}^\dagger)$ is a linear combination of annihilation and creation operators with $c$-dependent coefficients. We can build states by acting on the vacuum with the representation $\mathcal{O}(\vec{k})$,

$$|\mathcal{O}(\vec{k})\rangle \equiv \mathcal{O}(\vec{k})|0\rangle = \mathcal{N}(c)k^c|-\vec{k}\rangle, \tag{35}$$

where $\mathcal{N}(c)$ is a $c$-dependent coefficient. Ideally, these states should be normalizable up to an overall Dirac delta function coming from the normalization of single particle states $|\vec{k}\rangle$ which are normalized as

$$\langle \vec{k}'|\vec{k}\rangle = (2\pi)^3\delta^3(\vec{k}' - \vec{k}). \tag{36}$$

That is, defining $\Omega \equiv \int \frac{d^3k}{(2\pi)^3}\langle -\vec{k}|-\vec{k}\rangle$, we expect to find states normalized as

$$(\mathcal{O}, \mathcal{O}) = \frac{1}{\Omega}\int \frac{d^3k}{(2\pi)^3}\langle \mathcal{O}(\vec{k})|\mathcal{O}(\vec{k})\rangle = 1. \tag{37}$$

For late-time operators with purely imaginary scaling weight $c = i\rho$, which we will label by $H$ for heavy, since

$$|\mathcal{O}^H(\vec{k})\rangle = \mathcal{N}(i\rho)k^{i\rho}|-\vec{k}\rangle, \tag{38}$$

$$\langle \mathcal{O}^H(\vec{k})| = \mathcal{N}^*(i\rho)k^{-i\rho}\langle -\vec{k}|, \tag{39}$$

the expected normalization condition (37) is automatically normalized.

One case where real scaling weight $c = \mu$ occurs is for scalars with mass $m < \frac{3H}{2}$. These we will refer to as *light* fields and denote with label $L$. Here, with

$$|\mathcal{O}^L(\vec{k})\rangle = \mathcal{N}(\mu)k^\mu|-\vec{k}\rangle, \tag{40}$$

$$\langle \mathcal{O}^L(\vec{k})|\mathcal{O}^L(\vec{k}')\rangle = |\mathcal{N}(\mu)|^2 k^{2\mu}\langle -\vec{k}|-\vec{k}'\rangle \tag{41}$$

the factor $k^{2\mu}$ corresponds to the factor $|a|^{-(c^*+c)}$, that becomes problematic for real scaling weight in (31).

All is not lost on the real scaling weight front. We arrived at Equation (31) by assuming that the ket and bra states are associated to each other via the straight forward complex adjoint. This holds true for the principal series representations. For unitary representations with a real scaling weight $c \in \mathbb{R}$, the question of a well-defined inner product boils down to the question of what will be the well-defined adjoint? If the adjoint operation for $c \in \mathbb{R}$ involves a map that will cancel out the factor of $|a|^{-(c^*+c)}$, then we have a well-defined inner product and unitary irreducible representations with respect to that adjoint.

Luckily, an invertible map that does this job exists and has been well-established in mathematics literature. There is a map that assigns a representation $\tilde{\mathcal{O}}$ with $\tilde{c} = -c$ for each representation $\mathcal{O}$ with $c$. This is called the *intertwining map* [11]. It is a map between a function space $\mathcal{C}_\chi$ with functions of dimension $\Delta$ and a function space $\mathcal{C}_{\tilde{\chi}}$ with functions of dimension $\tilde{\Delta}$, such that

$$\Delta + \tilde{\Delta} = 3. \tag{42}$$

In general dimensions, the relation is $\Delta + \tilde{\Delta} = d$. This transformation is also called *the shadow transformation* in conformal field theory literature and the representations with dimensions $\tilde{\Delta}$ are referred to as *shadow transformations*. While $\chi = \{l, c\}$, the shadow representations[1] have $\tilde{\chi} = \{l, \tilde{c} = -c\}$.

In making use of any map, one needs to pay attention to the domain of the map. The normalized intertwining map and its inverse, which we will collectively denote by $G$, are well defined at different ranges of $c$ and act on the following function spaces [11]

$$G_{\chi} : \mathcal{C}_{\tilde{\chi}} \to \mathcal{C}_{\chi} \quad \text{such that} \quad Re(c) < 0 \tag{43}$$

$$G_{\tilde{\chi}} : \mathcal{C}_{\chi} \to \mathcal{C}_{\tilde{\chi}} \quad \text{such that} \quad Re(c) > 0, \tag{44}$$

such that

$$G_{\chi} G_{\tilde{\chi}} = G_{\tilde{\chi}} G_{\chi} = 1. \tag{45}$$

In $\vec{x}-$space the operator $G_{\chi}$ is related to the two-point function. We refer the reader to any one of the references [2,11,13] for the $\vec{x}-$space expression. In momentum space

$$G_{\chi}(k) = n(\chi) \frac{\Gamma(-c)}{(\frac{3}{2} + l + c - 1)\Gamma(\frac{3}{2} + c - 1)} \left(\frac{k^2}{2}\right)^c \sum_{s=0}^{l} K_{ls}(c) \Pi^{ls}(k). \tag{46}$$

The factors $\gamma_{\chi}$ and $n(\chi)$ are normalization factors related to each other as

$$\gamma_{\chi} = \frac{(-1)^l}{(2\pi)^{3/2}} 2^{\frac{3}{2}+c} n(\chi), \tag{47}$$

$\Pi^{ls}(k)$ are projection operators for whose details we refer to [2,11]. Here, we want to pay attention to the normalization of the intertwining operators. The explicit form of the coefficient $K_{ls}(c)$ works into the normalizability properties of the intertwining operators

$$K_{ls}(c) = (-1)^{l-s} \frac{\Gamma(\frac{3}{2} + c + s - 1)\Gamma(c + 2 - \frac{3}{2} - s)}{\Gamma(\frac{3}{2} + c + l - 1)\Gamma(c + 2 - \frac{3}{2} - l)}. \tag{48}$$

Note all the factors of Gamma functions involved in the definition of the intertwining operator. Gamma functions have poles whenever their argument is zero or takes on a negative integer value, that is, for $\Gamma(z)$, the poles are at $z = 0, -1, -2, \ldots$ The intertwining operator involves a bunch of Gamma functions that depend on the scaling weight and spin of the representation as well as the number of spatial dimensions. At certain combinations we will reach the poles of the Gamma function. Hence, in picking the normalization of the intertwining operator one needs to pay attention to these poles.

This means a different choice of normalization is appropriate for different values of $c$. This also ties into having different categories of representations when $c$ is real. The case of real scaling weight hosts three different categories of unitary irreducible representations, each of which have a different range of $c$. The range of $c$ affects the normalization of the intertwining operator and, hence, each category is equipped with a different intertwining operator, where the difference comes from the normalization. In general dimensions and for integer values of spin, these categories are [11]

$$
\begin{array}{lll}
\text{complementary series:} & \chi & \begin{array}{ll} l = 0 : & -\frac{d}{2} < c < \frac{d}{2}, \quad d \geq 2 \\[4pt] l = 1,2,3,\ldots : & 1 - \frac{d}{2} < c < \frac{d}{2} - 1, \quad d > 2 \end{array}
\end{array}
$$

$$
\begin{array}{lll}
\text{exceptional series:} & \begin{array}{l} \chi_{l_\sigma}^{-} \\ \chi_{l_\sigma}^{+} \end{array} & l : \quad \begin{array}{l} c = 1 - \frac{d}{2} - l - \sigma \\ c = \frac{d}{2} + l + \sigma - 1 \end{array} \\[10pt]
(l = 0,1,2,\ldots) & \chi_{l_\sigma}^{\prime -} & \\
(\sigma = 1,2,\ldots) & \chi_{l_\sigma}^{\prime +} & l + \sigma : \quad \begin{array}{l} c = 1 - \frac{d}{2} - l \\ c = \frac{d}{2} + l - 1 \end{array} \\[14pt]
\text{discrete series:} & \chi & l \qquad\quad \Delta = \frac{d}{2} + c \in \mathbb{Z} \quad d = 2,4
\end{array}
\tag{49}
$$

As we already mentioned in four spacetime dimensions, the exceptional series representations correspond to discrete series. For complementary series representations, the intertwining operator sets a similarity map. Under this operation, the two operators have the same trace, also referred to as the *character*, and therefore produce equivalent representations. In the case of discrete series, the two representations related by the intertwining map are not equivalent. The full list of intertwining operators for each category is provided in [11,13]. Here, we will only mention the ones as we make use of them, so as to be able to demonstrate their use without diverting the discussion too much.

For the complementary series representations in four spacetime dimensions, the range of the scaling weight for scalars and integer values of spin is [11]

$$
l = 0 : \quad -\frac{3}{2} < c < \frac{3}{2},
\tag{50}
$$

$$
l = 1,2,3,\ldots : \quad -\frac{1}{2} < c < \frac{1}{2},
\tag{51}
$$

and the appropriate choice of normalization is

$$
n^+(\chi) = \left( \frac{3}{2} + l + c - 1 \right) \frac{\Gamma\left(\frac{3}{2} + c - 1\right)}{\Gamma(-c)}
\tag{52}
$$

For scalars, $K_{00} = \Pi^{00} = 1$ and with the normalization (52) we have

$$
G^+_{\chi = \{0,c\}} = \left( \frac{k^2}{2} \right)^c \quad \text{where} \quad G^+_\chi : \mathcal{C}_{\tilde{\chi}} \to \mathcal{C}_\chi.
\tag{53}
$$

This intertwining operator works in the case of operators with a negative scaling weight, which we will denote by $\alpha^L(\vec{k})$, where $\alpha^L(\vec{k}) \in C_\chi$ as follows

$$
\alpha^L(\vec{k}) = G^+_\chi(k)\tilde{\alpha}^L(\vec{k}).
\tag{54}
$$

The inverse intertwining operator is

$$
G^+_{\tilde{\chi} = \{0,\tilde{c}\}} = \left( \frac{k^2}{2} \right)^{\tilde{c}} \quad \text{where} \quad G^+_{\tilde{\chi}} : \mathcal{C}_\chi \to \mathcal{C}_{\tilde{\chi}}
\tag{55}
$$

and this is well-defined for positive scaling weight. It acts on operators with positive scaling weight which we will denote by $\beta^L(\vec{k})$, such that $\beta^L(\vec{k}) \in \mathcal{C}_\chi$, as

$$
\tilde{\beta}^L(\vec{k}) = G^+_{\tilde{\chi}} \beta^L(\vec{k}).
\tag{56}
$$

In four spacetime dimensions, whenever the normalization (52) becomes ill defined, we reach discrete series representations. We refer to [30] for a very quick summary of discrete series representations and [13] for a more detailed review. The massless scalar

field in four spacetime dimensions has $c = \pm\frac{3}{2}$ and hence is categorized to host two representations from the category of discrete series representations with $\Delta = 0, 3$ which we will, respectively, denote as $\alpha^M(\vec{k})$ and $\beta^M(\vec{k})$. The intertwining operator normalized, as in the case of the complementary series category, becomes problematic for the representation with $c = -\frac{3}{2}$, $\Delta = 0$. This representation is identified to belong to $\chi_{01}^-$ with values $l = 0$, $\sigma = 1$. It lives in the function space $\mathcal{C}_{01}^-$, where the domain of the intertwining operator $G_{\chi_{01}^+}^+$ lies. In momentum space, this intertwining operator is given by [11]

$$G_{\chi_{01}^+}^+(k) = \left(\frac{k^2}{2}\right)^{\frac{3}{2}}, \quad \text{with} \quad G_{\chi_{01}^+}^+ : \mathcal{C}_{01}^- \to \mathcal{C}_{01}^+, \tag{57}$$

and normalizes the massless late-time operator $\alpha^M$ as follows

$$\tilde{\alpha}^M(\vec{k}) = G_{\chi_{01}^+}^+(k)\alpha^M(\vec{k}). \tag{58}$$

In general dimensions, this intertwining operator has the following expression [11]

$$G_{\chi_{l\sigma}^+}^+(k) = \left(\frac{k^2}{2}\right)^{l+\sigma+\frac{d}{2}-1} \sum_{s=0}^{l} \frac{(d+l+s+\sigma-3)!(\sigma+l-s)!}{(d+2l+\sigma-3)!\sigma!}(-1)^{l-s}\Pi^{ls}(k) \tag{59}$$

from which one can confirm that there are no poles in (57). The intertwining operator (55) can still be used in the normalization of the operator with $c = \frac{3}{2}$, $\Delta = 3$.

Note that it is not just the functional form of the intertwining operators $G_\chi^+$, $G_{\tilde{\chi}}^+$, $G_{\chi_{01}^+}^+$ that matter. One must pay attention to which function space they can act on.

The late-time operators $\mathcal{O}(\vec{x})$, obtained in free scalar theory, correspond to the unitary irreducible representations $(T_g^\chi f)(\vec{x})$ because they have scaling weights as expected from the scaling weights of the unitary irreducible representations of $SO(4,1)$ and they are normalizable with respect to the well-defined inner product, with the appropriate definition of the adjoint in each category recognized with respect to the scaling weight, as explicitly shown in [2].

To summarize, the unitary irreducible representations of $SO(4,1)$ fall into the following categories equipped with a different definition of the adjoint shown in Table 1.

**Table 1.** Categories of the representations of $SO(4,1)$ and the appropriate intertwining operators mentioned in the discussion.

| $c = i\rho$ | $(\mathcal{O}, \mathcal{O})$ | $\rho \in R^+$ | | **Principal** |
|---|---|---|---|---|
| $c \in R$ | $(\mathcal{O}, \tilde{\mathcal{O}})$ | $-\frac{3}{2} < c < \frac{3}{2}$ | for $c > 0$: $\tilde{\mathcal{O}} = G_{\tilde{\chi}}^+\mathcal{O}$ <br> for $c < 0$: $\mathcal{O} = G_\chi^+\tilde{\mathcal{O}}$ | Complementary |
| | | $\frac{3}{2} + c \in Z$ | for $c = -\frac{3}{2}$: $\tilde{\mathcal{O}} = G_{\chi_{01}^+}^+\mathcal{O}$ | Discrete |

In using the intertwining operators, one needs to take care of the sign of scaling weight for each late-time operator and the domains of the intertwining operators $G_\chi$ and $G_{\tilde{\chi}}$ to decide how to perform the shadow transformation in obtaining the adjoint operator $\tilde{\mathcal{O}}$. This is emphasized with explicit examples from the complementary series case in [2].

The inner product (29) for the function space on which the representations live does not contradict the Klein–Gordon inner product,

$$\left(F_{\vec{k}}, F_{\vec{k}'}\right)_{KG} \equiv -i \int d^3x \sqrt{|detg|}g^{0\mu}\left(F_{\vec{k}}^*\partial_\mu F_{\vec{k}'} - F_{\vec{k}'}\partial_\mu F_{\vec{k}}^*\right). \tag{60}$$

It is the mode functions that are normalized with respect to the Klein–Gordon inner product and it is straightforward to see that this remains true in the late time limit. Information

about the annihilation and creation operators do not work into the mode functions but they do work into the late-time operators and, hence, the representations they realize. The inner product for the representations is one that can work with the structure arising from the annihilation and creation operators and momentum or position dependence, while the Klein–Gordon inner product focuses on the time dependence. Readers interested in the connection between Klein–Gordon inner product and unitarity of the representations can also consider reference [25].

## 4. At the Late-Time Boundary

In this section, we give an example to each one of the categories of unitary irreducible representations we discussed in Section 3. Our examples are based on recognizing operators that realize these representations for free scalar fields at the late-time boundary. We refer to these operators as *late-time operators*. Following [2,3] we introduce these late-time operators and discuss their contribution to the late-time limit of two-point functions. Here, we work in terms of momentum modes and all the properties of the representations mentioned in 3 are encoded in the normalization of these operators. In a complementary way, it is possible to construct local operators that realize the unitary irreducible representations as well. Examples of this are [31,32], where the different properties of the different categories of unitary representations are encoded in the definition of the annihilation and creation operators (for instance, in [32], see the appearance of the intertwining operator in the commutation relation for complementary series annihilation and creation operators).

### 4.1. The Massive Scalar Field and Principal and Complementary Series Representations

Consider a free scalar field of various mass, on a fixed de Sitter metric

$$S = -\frac{1}{2} \int dt d^3x \sqrt{-g} \left[ g^{\mu\nu} \partial_\mu \phi \partial_\nu \phi + m^2 \phi^2 \right],$$
(61)

in the Poincaré patch with conformal time coordinate

$$ds^2 = \frac{-d\eta^2 + d\vec{x}^2}{H^2 |\eta|^2}.$$
(62)

In momentum space, the solution to mode functions that satisfy Bunch–Davies initial conditions, that is, the solutions that behave as if they were on Minkowski at very early times, involve Hankel functions.[2] These are Hankel functions of real order for light fields and of imaginary order for heavy fields, where lightness or heaviness of the field is determined in comparison to $\frac{3H}{2}$ in four spacetime dimensions. Therefore heavy and light scalars on frozen de Sitter, respectively, behave as follows

$$m > \frac{3}{2}H: \quad \phi^H(\vec{x}, \eta) = \int \frac{d^3k}{(2\pi)^3} \left[ |\eta|^{\frac{3}{2}} \tilde{H}_\rho^{(1)}(k|\eta|) a_{\vec{k}} + |\eta|^{\frac{3}{2}} \left( \tilde{H}_\rho^{(1)}(k|\eta|) \right)^* a_{-\vec{k}}^\dagger \right] e^{i\vec{k}\cdot\vec{x}}, \quad (63)$$

$$m < \frac{3}{2}H: \quad \phi^L(\vec{x}, \eta) = \int \frac{d^3k}{(2\pi)^3} \left[ |\eta|^{\frac{3}{2}} H_\nu^{(1)}(k|\eta|) a_{\vec{k}} + |\eta|^{\frac{3}{2}} \left( H_\nu^{(1)}(k|\eta|) \right)^* a_{-\vec{k}}^\dagger \right] e^{i\vec{k}\cdot\vec{x}}, \quad (64)$$

where $\tilde{H}_\rho^{(1)}(z) = e^{-\rho\pi/2} H_{i\rho}^{(1)}(z)$. In a wavefunction calculation for heavy fields, the difference between $\tilde{H}_\mu^{(1)}(u)$ and $H_{i\mu}^{(1)}(u)$ due to the numerical factor is not relevant but shortly we will talk about normalized late-time operators where this factor will work into the normalization.

The annihilation and creation operators $a_{\vec{k}}, a_{\vec{k}}^\dagger$ above satisfy the following commutation relations in our conventions

$$\left[ a_{\vec{k}}, a_{\vec{k}'}^\dagger \right] = (2\pi)^3 \delta^{(3)}(\vec{k} - \vec{k}').$$
(65)

Denoting the vacuum by $|0\rangle$ and a single particle state with momentum $\vec{k}$ by $|\vec{k}\rangle$, we work with states that are orthonormal

$$\langle \vec{k}|\vec{k}'\rangle = (2\pi)^3 \delta^{(3)}(\vec{k} - \vec{k}').\tag{66}$$

The annihilation and creation operators act on these states as follows

$$a_{\vec{k}}|0\rangle = 0, \quad a_{\vec{k}}^\dagger|0\rangle = |\vec{k}\rangle \text{ for } \forall \vec{k}.\tag{67}$$

In our notation, $\rho$ and $\nu$ are positive real parameters that carry information about the mass of the field

$$\rho^2 = \frac{m^2}{H^2} - \frac{9}{4}, \quad \nu^2 = \frac{9}{4} - \frac{m^2}{H^2}.\tag{68}$$

Notice that at a finite time $\eta \neq 0$, the time and momentum dependence of the scalar field behavior in (63) and (64) is interwoven. Yet in the late-time limit, as $\eta \to 0$, the time and momentum dependence factorizes and this involves two separate pieces that transform differently under a rescaling of coordinates. For the purpose of having a compact notation, defining the following labels for light and heavy fields

$$p = \{L, H\}, \quad \text{and} \quad \mu_p = \begin{cases} \mu_L = \nu \\ \mu_H = i\rho \end{cases}\tag{69}$$

the late-limit of the scalar field takes the form

$$\lim_{\eta \to 0} \phi^p(\vec{x}, \eta) = \int \frac{d^3k}{(2\pi)^3} \left[ |\eta|^{\frac{3}{2} - \mu_p} \alpha^p(\vec{k}) + |\eta|^{\frac{3}{2} + \mu_p} \beta^p(\vec{k}) \right] e^{i\vec{k}\cdot\vec{x}}.\tag{70}$$

We call $\alpha^p$ and $\beta^p$ the late-time operators. Note that so far we have not mentioned anything about normalization. Shortly we will discuss the connection between a field that is normalized with respect to Klein–Gordon normalization and late-time operators normalized with respect to appropriate inner products of the representation category they correspond to as we discussed in Section 3.

The late-time operators are composed of annihilation and creation operators and have momentum dependence as $k^{\pm \mu_p}$, where the plus sign applies for $\beta^p$ and minus sign for $\alpha^p$. These operators can be recognized as unitary irreducible representations of $SO(4,1)$ and they exist in other dimensions as well. One can confirm that they are unitary irreducible representations by checking their scaling dimensions and normalization properties, as was explicitly discussed in [2]. In the case of heavy scalars, the scaling dimensions $\Delta_{\mathcal{O}}$ are

$$m > \frac{3}{2}H : \quad \begin{aligned} \Delta_{\alpha^H} &= \tfrac{3}{2} - i\rho \\ \Delta_{\beta^H} &= \tfrac{3}{2} + i\rho. \end{aligned}\tag{71}$$

Notice that these scaling dimensions have purely imaginary scaling weights $c = \pm i\rho$. This is a defining property of the principal series representations. Moreover, the operators $\alpha^H$, $\beta^H$ are normalizable with respect to the principal series inner product, Equation (37) of Section 3. As such $\alpha^H$ and $\beta^H$ furnish an example to principal series representations. With appropriate normalization these operators are

$$\alpha_N^H(\vec{k}) = \sqrt{\rho\pi \sinh(\rho\pi)} \left[ -\frac{i}{\pi}\Gamma(i\rho)e^{-\rho\pi}a_{\vec{k}} + \frac{1}{\sinh(\rho\pi)\Gamma(1 - i\rho)}a_{-\vec{k}}^\dagger \right] \left(\frac{k}{2}\right)^{-i\rho}\tag{72}$$

$$\beta_N^H(\vec{k}) = \sqrt{\rho\pi \sinh(\rho\pi)} \left[ \frac{e^{\rho\pi}}{\sinh(\rho\pi)\Gamma(1 + i\rho)}a_{\vec{k}} + \frac{i}{\pi}\Gamma(-i\rho)a_{-\vec{k}}^\dagger \right] \left(\frac{k}{2}\right)^{i\rho}\tag{73}$$

where we follow the conventions of [3]. Similarly for light fields the scaling dimensions are

$$
m < \frac{3}{2}H: \quad \begin{array}{l} \Delta_{\alpha^L} = \frac{3}{2} - \nu \\[2mm] \Delta_{\beta^L} = \frac{3}{2} + \nu. \end{array}
\tag{74}
$$

These scaling dimensions have real scaling weights $c = \pm\nu$ and for masses that fall in the range $-\frac{3}{2} < c < \frac{3}{2}$ the late-time operators are normalizable with respect to the complementary series inner product as reviewed in Section 3 which involves shadow transformations. This range of real scaling weights only excludes the massless case. The appropriately normalized complementary series late-time operators are [3]

$$
0 < m < \frac{3}{2}H: \quad \begin{array}{l} \alpha_N^L(\vec{k}) = -i2^{\nu/2}\Big[a_{\vec{k}} - a_{-\vec{k}}^\dagger\Big]k^{-\nu} \\[3mm] \beta_N^L(\vec{k}) = 2^{-\nu/2}\Big[\frac{1+i\cot(\pi\nu)}{1-i\cot(\pi\nu)}a_{\vec{k}} + a_{-\vec{k}}^\dagger\Big]k^{\nu}. \end{array}
\tag{75}
$$

The complementary series case also involves the shadow operators that are obtained via the transformations in Equations (54) and (56) of Section 3. The normalized shadow operators and their scaling dimensions are [3]

$$
\tilde{\alpha}_N^L(\vec{k}) = -i2^{-\nu/2}\Big[a_{\vec{k}} - a_{-\vec{k}}^\dagger\Big]k^{\nu}, \quad \tilde{\Delta}_\alpha = \frac{3}{2} + \nu
\tag{76}
$$

$$
\tilde{\beta}_N^L(\vec{k}) = 2^{\nu/2}\Big[\frac{1 + i\cot(\pi\nu)}{1 - i\cot(\pi\nu)}a_{\vec{k}} + a_{-\vec{k}}^\dagger\Big]k^{-\nu}, \quad \tilde{\Delta}_\beta = \frac{3}{2} - \nu
\tag{77}
$$

Note that the operators $\tilde{\alpha}_N^L$ and $\beta_N^L$ have the same value of the scaling dimension, the same observation holds for $\tilde{\beta}_N^L$ and $\alpha_N^L$. However, comparing the terms in the square parenthesis of Equations (75) and (76) shows that $\beta_N^L$ is not the same operator as $\tilde{\alpha}_N^L$, similarly $\alpha_N^L$ is not the same as $\tilde{\beta}_N^L$. This is why we do not refer to $\alpha_N^L$ and $\beta_N^L$ as shadows of each other.

In inflationary calculations on the correlation functions of scalars, a useful convention is to calculate the correlation functions of the Fourier modes of the canonically quantized field operator, the $\varphi_{\vec{k}}(\eta)$ defined via

$$
\phi_{KG}(\vec{x}, \eta) = \int \frac{d^3k}{(2\pi)^3} e^{i\vec{k}\cdot\vec{x}} \varphi_{\vec{k}}(\eta)
\tag{78}
$$

where the mode functions $F_{\vec{k}}(\vec{x}, \eta) = \varphi_{\vec{k}}(\eta)e^{i\vec{k}\cdot\vec{x}}$, have been normalized with respect to the Klein–Gordon norm (60). If we take a step back, the full set of canonically conjugate pair of variables involve a conjugate momentum operator as well,

$$
\pi_{KG}(\vec{x}, \eta) = \int \frac{d^3k}{(2\pi)^3} e^{i\vec{k}\cdot\vec{x}} \pi_{\vec{k}}(\eta).
\tag{79}
$$

This pair of conjugate operators have the nontrivial commutation relation

$$
\big[\varphi_{\vec{k}}(\eta), \pi_{\vec{k}'}(\eta)\big] = i(2\pi)^3 \delta^{(3)}(\vec{k} + \vec{k}').
\tag{80}
$$

We can also write the late-time operators $\{\alpha_N, \beta_N\}$ in terms of canonical field and momentum modes, $\{\varphi_{\vec{k}}, \pi_{\vec{k}}\}$. Denoting $\eta_0 \equiv 0$ and defining

$$
\lim_{\eta \to \eta_0 = 0} \phi_{KG}^p(\vec{x}, \eta) = \int \frac{d^3k}{(2\pi)^3} e^{i\vec{k}\cdot\vec{x}} \varphi_{\vec{k}}^{p,lt}(\eta_0),
\tag{81}
$$

$$
\lim_{\eta \to \eta_0 = 0} \pi_{KG}^p(\vec{x}, \eta) = \int \frac{d^3k}{(2\pi)^3} e^{i\vec{k}\cdot\vec{x}} \pi_{\vec{k}}^{p,lt}(\eta_0)
\tag{82}
$$

from (70), we have

$$\varphi_{\vec{k}}^{p,lt}(\eta_0) = |\eta_0|^{\frac{3}{2}-\mu_p}\alpha^p(\vec{k}) + |\eta|^{\frac{3}{2}+\mu_p}\beta^p(\vec{k}), \tag{83}$$

$$\pi_{\vec{k}}^{p,lt}(\eta_0) = -\left(\frac{3}{2}-\mu_p\right)\frac{|\eta_0|^{-\frac{3}{2}-\mu_p}}{H^2}\alpha^p(\vec{k}) - \left(\frac{3}{2}+\mu_p\right)\frac{|\eta_0|^{-\frac{3}{2}+\mu_p}}{H^2}\beta^p(\vec{k}). \tag{84}$$

So far, everything looks in unison for the light and heavy fields. However, reconsidering these expressions in terms of the normalized late-time operators highlights differences in the coefficients [3]. For light fields we have

$$\varphi_{\vec{k}}^{L,lt}(\eta_0) = \frac{\sqrt{\pi}}{2}H\left[|\eta_0|^{\frac{3}{2}-\nu}\frac{\Gamma(\nu)}{\pi}2^{\nu/2}\alpha_N^L(\vec{k}) + |\eta_0|^{\frac{3}{2}+\nu}\frac{1-i\cot(\pi\nu)}{2^{\nu/2}\Gamma(\nu+1)}\beta_N^L(\vec{k})\right], \tag{85a}$$

$$\pi_{\vec{k}}^{L,lt}(\eta_0) = -\frac{\sqrt{\pi}}{2H}\left[|\eta_0|^{-\frac{3}{2}-\nu}\left(\frac{3}{2}-\nu\right)\frac{\Gamma(\nu)}{\pi}2^{\nu/2}\alpha_N^L(\vec{k})\right.$$
$$\left. + |\eta_0|^{-\frac{3}{2}+\nu}\left(\frac{3}{2}+\nu\right)\frac{1-i\cot(\pi\nu)}{2^{\nu/2}\Gamma(\nu+1)}\beta_N^L(\vec{k})\right]; \tag{85b}$$

whereas for heavy fields, the relation is

$$\varphi_{\vec{k}}^{H,lt}(\eta_0) = \frac{H}{2\sqrt{\rho\sinh(\rho\pi)}}\left[|\eta_0|^{\frac{3}{2}-i\rho}e^{\pi\rho/2}\alpha_N^H(\vec{k}) + |\eta_0|^{\frac{3}{2}+i\rho}e^{-\pi\rho/2}\beta_N^H(\vec{k})\right], \tag{86a}$$

$$\pi_{\vec{k}}^{H,lt}(\eta_0) = -\frac{1}{2H\sqrt{\rho\sinh(\rho\pi)}}\left[|\eta_0|^{-\frac{3}{2}-i\rho}\left(\frac{3}{2}-i\rho\right)e^{\pi\rho/2}\alpha_N^H(\vec{k})\right.$$
$$\left. + |\eta_0|^{-\frac{3}{2}+i\rho}\left(\frac{3}{2}+i\rho\right)e^{-\pi\rho/2}\beta_N^H(\vec{k})\right]. \tag{86b}$$

In observations, the physical measurements probe $\varphi_{\vec{k}}^{p,lt}(\eta_0)$ and $\pi_{\vec{k}}^{p,lt}(\eta_0)$ as opposed to the late-time operators. The above relations can be inverted to express $\alpha_N^p(\vec{k})$ and $\beta_N^p(\vec{k})$ in terms of $\varphi_{\vec{k}}^{p,lt}(\eta_0)$, $\pi_{\vec{k}}^{p,lt}(\eta_0)$. For example, the heavy late-time operators can be written in terms of the heavy field momentum modes as

$$\alpha_N^H(\vec{k}) = -i\sqrt{\frac{\sinh(\rho\pi)}{\rho}}\left[\frac{\varphi_{\vec{k}}^{H,lt}}{H}|\eta_0|^{-\frac{3}{2}+i\rho}\left(\frac{3}{2}+i\rho\right)e^{-\pi\rho/2} + H\pi_{\vec{k}}^{H,lt}|\eta_0|^{\frac{3}{2}+i\rho}e^{-\pi\rho/2}\right], \tag{87}$$

$$\beta_N^H(\vec{k}) = i\sqrt{\frac{\sinh(\rho\pi)}{\rho}}\left[\frac{\varphi_{\vec{k}}^{H,lt}}{H}|\eta_0|^{-\frac{3}{2}-i\rho}\left(\frac{3}{2}-i\rho\right)e^{\pi\rho/3} + H\pi_{\vec{k}}^{H,lt}|\eta_0|^{\frac{3}{2}-i\rho}e^{\pi\rho/2}\right]. \tag{88}$$

where we suppressed the $\eta_0$ dependence of $\varphi_{\vec{k}}^{p,lt}(\eta_0)$, $\pi_{\vec{k}}^{p,lt}(\eta_0)$ for ease of notation.

Since the late-time operators are built out of annihilation and creation operators, they inherit a nontrivial commutation relation coming from the commutation properties of annihilation and creation operators. This differs in terms of the coefficients in the case of principal and complementary series

$$\left[\beta_N^H(\vec{k}), \alpha_N^H(\vec{k'})\right] = -2\sinh(\rho\pi)(2\pi)^3\delta^{(3)}\left(\vec{k}+\vec{k'}\right), \tag{89}$$

$$\left[\alpha_N^L(\vec{k}), \beta_N^L(\vec{k'})\right] = \frac{-2i}{1-i\cot(\nu\pi)}(2\pi)^3\delta^{(3)}\left(\vec{k}+\vec{k'}\right). \tag{90}$$

This resembles the commutation of field and conjugate momentum operators.

Note that we can also invert Equations (72), (73) and (75) to write the annihilation and creation operators in terms of the late-time operators. In the case of light fields, this relation is

$$a_{\vec{k}} = \frac{i\cot(\nu\pi) - 1}{2\cot(\nu\pi)}\left[\frac{k^\nu}{2^{\nu/2}}\alpha_N^L(\vec{k}) + i\frac{2^{\nu/2}}{k^\nu}\beta_N^L(\vec{k})\right] \tag{91}$$

$$a_{-\vec{k}}^\dagger = \frac{1 + i\cot(\nu\pi)}{i2^{1+\nu/2}}k^\nu\alpha_N^L(\vec{k}) + \frac{1 - i\cot(\nu\pi)}{2^{1-\nu/2}}k^{-\nu}\beta_N^L(\vec{k}) \tag{92}$$

whereas in the case of heavy fields, it is

$$a_{\vec{k}} = \frac{1}{2}\left[\frac{\sqrt{\rho\pi\sinh(\rho\pi)}}{\sinh^2(\rho\pi)\Gamma(1 - i\rho)}\left(\frac{k}{2}\right)^{-i\rho}\beta_N^H(\vec{k}) - \frac{i}{\pi}\Gamma(-i\rho)\sqrt{\frac{\rho\pi}{\sinh\rho\pi}}\left(\frac{k}{2}\right)^{i\rho}\alpha_N^H(\vec{k})\right] \tag{93}$$

$$a_{-\vec{k}}^\dagger = \frac{1}{2}\left[\frac{i}{\pi}\sqrt{\frac{\rho\pi}{\sinh(\rho\pi)}}\Gamma(i\rho)e^{-\rho\pi}\left(\frac{k}{2}\right)^{-i\rho}\beta_N^H(\vec{k}) + \frac{\sqrt{\rho\pi\sinh(\rho\pi)}}{\sinh^2(\rho\pi)\Gamma(1 + i\rho)}e^{\rho\pi}\left(\frac{k}{2}\right)^{i\rho}\alpha_N^H(\vec{k})\right]. \tag{94}$$

By acting on the $SO(4,1)$ invariant vacuum state that is annihilated by annihilation operators with these late-time operators, we can build single particle states

$$|\alpha_N^p(\vec{k})\rangle \equiv \alpha_N^p(\vec{k})|0\rangle, \tag{95}$$

$$|\beta_N^p(\vec{k})\rangle \equiv \beta_N^p(\vec{k})|0\rangle. \tag{96}$$

We noted the difference between $\alpha_N^L$ and $\beta_N^L$, even though their dimensions resemble the dimensions of operators related by shadow transformations. At the level of states, the states built from $\beta_N^L$ do match the states built from $\tilde{\alpha}_N^L$ up to an overall factor of $i$. This is also true in comparing the states built from $\alpha_N^L$ and $\tilde{\beta}_N^L$,

$$\alpha_N^L(\vec{k})|0\rangle = i\tilde{\beta}_N^L|0\rangle = i2^{\nu/2}k^{-\nu}|-\vec{k}\rangle, \tag{97}$$

$$\beta_N^L(\vec{k})|0\rangle = -i\tilde{\alpha}_N^L(\vec{k})|0\rangle = 2^{-\nu/2}k^\nu|-\vec{k}\rangle. \tag{98}$$

While at the level of operators $\alpha_N^L$ and $\beta_N^L$ are not shadows of each other, at the level of states they do give rise to states that can be recognized as shadows of each other.

Two Point Functions of Principal and Complementary Series Late-Time Operators

Now let us consider the two-point functions. Our notation will be such that

$$\langle\mathcal{O}(\vec{k})\mathcal{O}(\vec{k}')\rangle = \langle 0|\mathcal{O}(\vec{k})\mathcal{O}(\vec{k}')|0\rangle. \tag{99}$$

The two-point functions of the late-time operators themselves have the following momentum dependence [3], as also summarized in [33]. For the case of principal series representations

$$\langle\alpha_N^H(\vec{k})\alpha_N^H(\vec{k}')\rangle = -\frac{\Gamma(1 + i\rho)}{\Gamma(1 - i\rho)}e^{-\rho\pi}(2\pi)^3\delta^{(3)}(\vec{k} + \vec{k}')\left(\frac{k}{2}\right)^{-2i\rho},$$

$$= -e^{2i\gamma_\rho - \rho\pi}(2\pi)^3\delta^{(3)}(\vec{k} + \vec{k}')\left(\frac{k}{2}\right)^{-2i\rho} \tag{100a}$$

$$\langle\beta_N^H(\vec{k})\beta_N^H(\vec{k}')\rangle = i\rho\frac{\Gamma(-i\rho)}{\Gamma(1 + i\rho)}e^{\rho\pi}(2\pi)^d\delta^{(d)}(\vec{k} + \vec{k}')\left(\frac{k}{2}\right)^{2i\rho} \tag{100b}$$

$$= -e^{-2i\gamma_\rho + \rho\pi}(2\pi)^3\delta^{(3)}(\vec{k} + \vec{k}')\left(\frac{k}{2}\right)^{2i\rho}$$

$$\langle\alpha_N^H(\vec{k})\beta_N^H(\vec{k}')\rangle = e^{-\rho\pi}(2\pi)^3\delta^{(3)}(\vec{k} + \vec{k}'), \tag{100c}$$

$$\langle\beta_N^H(\vec{k})\alpha_N^H(\vec{k}')\rangle = e^{\rho\pi}(2\pi)^3\delta^{(3)}(\vec{k} + \vec{k}'). \tag{100d}$$

In the case of complementary series representations, the list is as follows

$$\langle \alpha_N^L(\vec{k})\alpha_N^L(\vec{k}')\rangle = 2^\nu k^{-2\nu}(2\pi)^3\delta^{(3)}(\vec{k}+\vec{k}'), \tag{101a}$$

$$\langle \beta_N^L(\vec{k})\beta_N^L(\vec{k}')\rangle = \frac{k^{2\nu}}{2^\nu}\frac{1+i\cot(\pi\nu)}{1-i\cot(\pi\nu)}(2\pi)^3\delta^{(3)}(\vec{k}+\vec{k}'), \tag{101b}$$

$$\langle \alpha_N^L(\vec{k})\beta_N^L(\vec{k}')\rangle = -i(2\pi)^3\delta^{(3)}(\vec{k}+\vec{k}') \tag{101c}$$

$$\langle \beta_N^L(\vec{k})\alpha_N^L(\vec{k}')\rangle = i\frac{1+i\cot(\pi\nu)}{1-i\cot(\pi\nu)}(2\pi)^3\delta^{(3)}(\vec{k}+\vec{k}'). \tag{101d}$$

Lastly, we would like to discuss how these late-time operators contribute to the late-time two-point functions. The relations in Equations (83) and (84) we introduced in the canonical quantization picture make the comparison easy. As also checked by wavefunction calculations [3], the two-point functions at late-time are related as follows for the principal series representations

$$\langle \varphi_{\vec{k}}^{H,lt}\varphi_{\vec{k}'}^{H,lt}\rangle = \frac{H^2|\eta_0|^3}{4\rho\sinh(\rho\pi)}\left[|\eta_0|^{-2i\rho}e^{\rho\pi}\langle \alpha_N^H(\vec{k})\alpha_N^H(\vec{k}')\rangle + |\eta_0|^{2i\rho}e^{-\rho\pi}\langle \beta_N^H(\vec{k})\beta_N^H(\vec{k}')\rangle\right.$$

$$\left. + \langle \alpha_N^H(\vec{k})\beta_N^H(\vec{k}')\rangle + \langle \beta_N^H(\vec{k})\alpha_N^H(\vec{k}')\rangle\right] \tag{102}$$

$$\langle \pi_{\vec{k}}^{H,lt}\pi_{\vec{k}'}^{H,lt}\rangle = \frac{|\eta_0|^{-3}}{4\rho\sinh(\rho\pi)H^2}\left\{\left(\frac{3}{2}+i\rho\right)^2|\eta_0|^{2i\rho}e^{-\rho\pi}\langle \beta_N^H(\vec{k})\beta_N^H(\vec{k}')\rangle\right. \tag{103}$$

$$+ \left(\frac{3}{2}-i\rho\right)^2|\eta_0|^{-2i\rho}e^{\rho\pi}\langle \alpha_N^H(\vec{k})\alpha_N^H(\vec{k}')\rangle$$

$$\left. + \left(\frac{9}{4}+\rho^2\right)\left[\langle \alpha_N^H(\vec{k})\beta_N^H(\vec{k}')\rangle + \langle \beta_N^H(\vec{k})\alpha_N^H(\vec{k}')\rangle\right]\right\}. \tag{104}$$

The explicit momentum dependence of these two-point functions are

$$\langle \varphi_{\vec{k}}^{H,lt}\varphi_{\vec{k}'}^{H,lt}\rangle = \frac{(2\pi)^3H^2|\eta_0|^3}{4\rho\sinh(\rho\pi)}\delta^{(3)}(\vec{k}+\vec{k}')\times$$

$$\left[2\cosh(\rho\pi) - e^{-2i\gamma_\rho}\left(\frac{k|\eta_0|}{2}\right)^{2i\rho} - e^{2i\gamma_\rho}\left(\frac{k|\eta_0|}{2}\right)^{-2i\rho}\right] \tag{105}$$

$$\langle \pi_{\vec{k}}^{H,lt}\pi_{\vec{k}'}^{H,lt}\rangle = \frac{(2\pi)^3}{4\rho\sinh(\rho\pi)|\eta_0|^3H^2}\delta^{(3)}(\vec{k}+\vec{k}')\left[\left(\frac{9}{4}+\rho^2\right)2\cosh(\rho\pi)\right.$$

$$\left. - \left(\frac{3}{2}+i\rho\right)^2e^{-2i\gamma_\rho}\left(\frac{k|\eta_0|}{2}\right)^{2i\rho} - \left(\frac{3}{2}-i\rho\right)^2e^{2i\gamma_\rho}\left(\frac{k|\eta_0|}{2}\right)^{-2i\rho}\right]. \tag{106}$$

For the complementary series representations, the two-point functions of the canonically quantized field and momenta are related to the two-point functions of the late-time operators as

$$\langle \varphi_{\vec{k}}^{L,lt} \varphi_{\vec{k}'}^{L,lt} \rangle = \frac{\pi}{4} H^2 |\eta_0|^3 \left\{ \frac{2^\nu \Gamma^2(\nu)}{\pi^2 |\eta_0|^{2\nu}} \langle \alpha_N^L(\vec{k}) \alpha_N^L(\vec{k}') \rangle \right.$$

$$+ \frac{(1 - i \cot(\nu\pi))^2}{2^\nu \Gamma^2(1+\nu)} |\eta_0|^{2\nu} \langle \beta_N^L(\vec{k}) \beta_N^L(\vec{k}') \rangle$$

$$\left. + \frac{1 - i \cot(\nu\pi)}{\nu\pi} \left[ \langle \beta_N^L(\vec{k}) \alpha_N^L(\vec{k}') \rangle + \langle \alpha_N^L(\vec{k}) \beta_N^L(\vec{k}') \rangle \right] \right\}, \tag{107}$$

$$\langle \pi_{\vec{k}}^{L,lt} \pi_{\vec{k}'}^{L,lt} \rangle = \frac{\pi}{4} \frac{1}{|\eta_0|^3 H^2} \left\{ \frac{1 - i \cot(\nu\pi)}{\nu\pi} \left( \frac{3}{2} + \nu \right) \left( \frac{3}{2} - \nu \right) \left[ \langle \beta_N^L(\vec{k}) \alpha_N^L(\vec{k}') \rangle \right. \right.$$

$$\left. + \langle \alpha_N^L(\vec{k}) \beta_N^L(\vec{k}') \rangle \right] + \frac{2^\nu \Gamma^2(\nu)}{\pi^2 |\eta_0|^{2\nu}} \left( \frac{3}{2} - \nu \right)^2 \langle \alpha_N^L(\vec{k}) \alpha_N^L(\vec{k}') \rangle$$

$$\left. + \frac{(1 - i \cot(\nu\pi))^2 |\eta_0|^{2\nu}}{2^\nu \Gamma^2(\nu+1)} \left( \frac{3}{2} + \nu \right)^2 \langle \beta_N^L(\vec{k}) \beta_N^L(\vec{k}') \rangle \right\}; \tag{108}$$

where the explicit momentum dependence is

$$\langle \varphi_{\vec{k}}^{L,lt} \varphi_{\vec{k}'}^{L,lt} \rangle = \frac{\pi (2\pi)^3}{4} H^2 |\eta_0|^3 \delta^{(3)}(\vec{k} + \vec{k}') \times$$

$$\left\{ \frac{\Gamma^2(\nu)}{\pi^2} \left( \frac{k|\eta_0|}{2} \right)^{-2\nu} + \frac{(1 + \cot^2(\nu\pi))}{\Gamma^2(1+\nu)} \left( \frac{k|\eta_0|}{2} \right)^{2\nu} - \frac{2}{\nu\pi} \cot(\pi\nu) \right\}, \tag{109}$$

$$\langle \pi_{\vec{k}}^{L,lt} \pi_{\vec{k}'}^{L,lt} \rangle = \frac{\pi}{4} \frac{(2\pi)^3}{|\eta_0|^3 H^2} \delta^{(3)}(\vec{k} + \vec{k}') \left[ \frac{\Gamma^2(\nu)}{\pi^2} \left( \frac{3}{2} - \nu \right)^2 \left( \frac{k|\eta_0|}{2} \right)^{-2\nu} \right.$$

$$\left. + \frac{1 + \cot^2(\nu\pi)}{\Gamma^2(\nu+1)} \left( \frac{3}{2} + \nu \right)^2 \left( \frac{k|\eta_0|}{2} \right)^{2\nu} - \frac{2 \cot(\nu\pi)}{\nu\pi} \left( \frac{3}{2} + \nu \right) \left( \frac{3}{2} - \nu \right) \right]. \tag{110}$$

*4.2. Massless Scalar and the Discrete Series Representations*

The massless scalar solution

$$m = 0: \quad \phi^L(\vec{x}, \eta) = \int \frac{d^3 k}{(2\pi)^3} \left[ |\eta|^{\frac{3}{2}} H_{\frac{3}{2}}^{(1)}(k|\eta|) a_{\vec{k}} + |\eta|^{\frac{3}{2}} \left( H_{\frac{3}{2}}^{(1)}(k|\eta|) \right)^* a_{-\vec{k}}^\dagger \right] e^{i\vec{k}\cdot\vec{x}}, \tag{111}$$

is the solution which accommodates two late-time operators with weights $c = \pm \frac{3}{2}$. This puts the massless scalar exactly at the values that fall just out of the category of complementary series. The late-time operator with scaling weight $c_\alpha = -\frac{3}{2}$, scaling dimension $\Delta_\alpha = 0$ is

$$\alpha^M(\vec{k}) = -\frac{i}{\pi} \Gamma\left( \frac{3}{2} \right) N_\alpha^M \left[ a_{\vec{k}} - a_{-\vec{k}}^\dagger \right] \left( \frac{k}{2} \right)^{-\frac{3}{2}}. \tag{112}$$

Notice that for this operator, the normalization of the intertwining operator $G_{\chi=0, -\frac{3}{2}}^+$ of Section 3, Equation (53) corresponds to

$$n^+\left( \chi = 0, -\frac{3}{2} \right) = -1 \frac{\Gamma(-1)}{\Gamma(\frac{3}{2})}. \tag{113}$$

The Gamma function $\Gamma(-1)$ has a pole and so we cannot normalize this operator with the complementary series inner product. The inner product that applies here is the one with the intertwining operator (57) of Section 3

$$G^{+}_{\chi^{+}_{01}}(k) = \left(\frac{k^2}{2}\right)^{\frac{3}{2}}, \tag{114}$$

and thus the normalized late-time operator of dimension $\Delta = 0$ is

$$\alpha_N^M(\vec{k}) = -i2^{3/4}\left[a_{\vec{k}} - a^{\dagger}_{-\vec{k}}\right]k^{-3/2}. \tag{115}$$

The second late-time operator has dimension $\Delta = 3$ and prior to normalization is of the form

$$\beta^M(\vec{k}) = \frac{N_\beta^M}{\Gamma(\frac{3}{2}+1)}\left[a_{\vec{k}} + a^{\dagger}_{-\vec{k}}\right]\left(\frac{k}{2}\right)^{\frac{3}{2}}. \tag{116}$$

The intertwining operation in equation (56) is applicable to this operator

$$\tilde{\beta}^M(\vec{k}) = G^{+}_{\tilde{\chi}=\{0,-\frac{3}{2}\}}(\vec{k})\beta^M(\vec{k}), \tag{117}$$

and its normalized form is

$$\beta_N^M(\vec{k}) = 2^{-\frac{3}{4}}\left[a_{\vec{k}} + a^{\dagger}_{-\vec{k}}\right]k^{\frac{3}{2}}. \tag{118}$$

The commutation relation between the massless late-time operators is

$$\left[\beta_N^M(\vec{k}), \alpha_N^M(\vec{k}')\right] = 2i(2\pi)^3\delta^{(3)}\left(\vec{k}+\vec{k}'\right). \tag{119}$$

In this case, the relation between the late-time field and conjugate momentum modes and the late-time operators are as follows

$$\varphi_{\vec{k}}^{M,lt}(\eta_0) = H\left[\frac{2^{1/4}}{3}|\eta_0|^3\beta_N^M(\vec{k}) + \frac{1}{2^{5/4}}\alpha_N^M(\vec{k})\right] \tag{120a}$$

$$\pi_{\vec{k}}^{M,lt}(\eta_0) = -H^{-1}2^{1/4}\beta_N^M(\vec{k}). \tag{120b}$$

Note that a special feature of this case is that conjugate momentum modes track solely $\beta_N^M(\vec{k})$ among the late-time operators.

As the scaling weight depends on the dimension, in general dimensions, some of the massless scalar solutions can be recognized as part of the exceptional series category. Following [13], in four dimensions the exceptional series category is considered equivalent to a discrete series category and we recognize $\alpha_N^M$ and $\beta_N^M$ to belong to discrete series representations, which also agrees with [32]. Other examples in this category are the non-zero spin fields in Higher Spin theory on de Sitter [34] and the spin$-\frac{3}{2}$ and $\frac{5}{2}$ fermions [35]. Another example of identifying discrete series representations comes from modified gravity literature [36], where the tensor degrees f freedom at a perturbative level are recognized to belong to discrete series representations of the de Sitter group.

Two Point Functions of Discrete Series Late-Time Operators

The two-point functions of the scalar discrete series late-time operators are

$$\langle \alpha_N^M(\vec{k}) \alpha_N^M(\vec{k'}) \rangle = \frac{2^{3/2}}{k^3}(2\pi)^3 \delta^{(3)}\left(\vec{k}+\vec{k'}\right) \tag{121a}$$

$$\langle \alpha_N^M(\vec{k}) \beta_N^M(\vec{k'}) \rangle = -i(2\pi)^3 \delta^{(3)}\left(\vec{k}+\vec{k'}\right) \tag{121b}$$

$$\langle \beta_N^M(\vec{k}) \alpha_N^M(\vec{k'}) \rangle = i(2\pi)^3 \delta^{(3)}\left(\vec{k}+\vec{k'}\right) \tag{121c}$$

$$\langle \beta_N^M(\vec{k}) \beta_N^M(\vec{k'}) \rangle = \frac{k^3}{2^{3/2}}(2\pi)^3 \delta^{(3)}\left(\vec{k}+\vec{k'}\right) \tag{121d}$$

Once again, we note how these late-time operator correlation functions contribute to the two-point functions of field and conjugate momenta operators at the late-time

$$\langle \varphi_{\vec{k}}^{M,lt} \varphi_{\vec{k'}}^{M,lt} \rangle = H^2 |\eta_0|^3 \left\{ \frac{|\eta_0|^{-3}}{2^{5/2}} \langle \alpha_N^{M,lt} \alpha_N^{M,lt} \rangle + \frac{\sqrt{2}}{9} |\eta_0|^3 \langle \beta_N^{M,lt} \beta_N^{M,lt} \rangle \right.$$
$$\left. + \frac{1}{6} \left[ \langle \beta_N^M \alpha_N^M \rangle + \langle \alpha_N^M \beta_N^M \rangle \right] \right\} \tag{122a}$$

$$\langle \pi_{\vec{k}}^{M,lt} \pi_{\vec{k'}}^{M,lt} \rangle = \frac{\sqrt{2}}{H^2} \langle \beta_N^M \beta_N^M \rangle. \tag{122b}$$

It is interesting to note that in this case the $\beta_N^M$ two-point function alone completely determines the conjugate momentum two-point function, as expected from the relation between the modes and late-time operators. The momentum dependence of these two-point functions is

$$\langle \varphi_{\vec{k}}^{M,lt} \varphi_{\vec{k'}}^{M,lt} \rangle = (2\pi)^3 H^2 \left[ \frac{|\eta_0|^6 k^3}{18} + \frac{k^{-3}}{2} \right] \delta^{(3)}\left(\vec{k}+\vec{k'}\right), \tag{123a}$$

$$\langle \pi_{\vec{k}}^{M,lt} \pi_{\vec{k'}}^{M,lt} \rangle = \frac{k^3}{2H^2}(2\pi)^3 \delta^{(3)}\left(\vec{k}+\vec{k'}\right). \tag{123b}$$

## 5. Wavefunction Discussions

Wavefunction techniques were initially introduced to address questions related to quantum gravity and quantum cosmology [37]. Recently, early universe studies have started consulting this technique more often then before. One example is in adressing inflationary perturbations in the regime where non-Gaussianities become important on the tail of the probability distribution and in–in methods fail [38]. Another similar example is its use in studying the infrared effects in presence of a quartic self interaction of a light scalar field on de Sitter [39]. Wavefunction methods also come in handy when discussing properties of interactions, especially for comparing the conditions on allowed interactions between the cases of a zero and positive cosmological constant. Each of the non-exhaustive list of references [40–43], has initiated a different direction in addressing the unitarity of interactions and bulk time evolution by studying the properties of the de Sitter wavefunction and its more general version for Friedmann Le Maitre Robertson Walker spacetime. Ref. [44] provides a short review of the recent progresses on this direction in terms of comparing restrictions on flat space physics, due to Lorentz invariance, locality, unitarity and causality that manifest themselves as properties of the S-matrix, and their counterpart in the case of a positive cosmological constant in terms of how the properties of the de Sitter wavefunction become restricted. We find reference [45] to be a pedagogical introduction to this technique and we will follow it in our summary below. Reference [46] is another pedagogical summary of the subject.

Earlier advances towards reaching this level of rigour and usefulness of the wavefunction technique in the case of a positive cosmological constant were established by studying the case of the massless scalar field [47] and the late-time behaviour of the wavefunction in the case of interacting light fields of various spin (namely scalars and gauge fields) on a de Sitter background [48].

Perhaps what motivated the study of the wavefunction in the case of a positive cosmological constant was the interpretation of [49] for holography with a positive cosmological constant in terms of a wavefunction description. This work and earlier works that followed its lead saw this holographic description as a complementary means to address inflationary calculations [50], which could eventually lead to a better interpretation of inflationary observations [51], such as the observed suppression of tensor degrees of freedom over scalars and the tilt in the power spectrum. Along the way, the wavefunction techniques have been employed in deriving consistency conditions on inflationary correlation functions [52]. These techniques also gave rise to discussions on studying the deviations from scale invariance exhibited in the inflationary power spectra in terms of renormalization group flows [53–56]. This discussion has led to classifying inflationary potentials into universality classes [57]. In a different direction, today, renormalization group techniques are even used to investigate interactions that lead to divergences with time within perturbative quantum field theory on de Sitter [58]. With this insight, let us now review the de Sitter wavefunction.

The wavefunction is a complex valued functional of the field eigenvalues $\phi(\vec{x}, \eta)$ and time

$$\Psi[\phi(\vec{x}, \eta), \eta]; \tag{124}$$

where the eigenstates $\{|\phi(\vec{x}, \eta)\rangle\}$ of the field operator $\hat{\phi}(\vec{x}, \eta)$,

$$\hat{\phi}(\vec{x}, \eta)|\phi(\vec{x}, \eta)\rangle = \phi(\vec{x}, \eta)|\phi(\vec{x}, \eta)\rangle, \tag{125}$$

form a complete basis at any fixed time $\eta$. The wavefunctional is defined as

$$\Psi[\phi(\vec{x}, \eta), \eta] \equiv \langle\phi(\vec{x}, \eta)|\Psi(\eta)\rangle, \tag{126}$$

An arbitary quantum mechanical state $|\Psi(\eta)\rangle$ at any given time can be expressed in terms of the basis $\{|\phi(\vec{x}, \eta)\rangle\}$ as follows

$$|\Psi(\eta)\rangle = \int \mathcal{D}\phi(\vec{x}, \eta)|\phi(\vec{x}, \eta)\rangle\langle\phi(\vec{x}, \eta)|\Psi(\eta)\rangle. \tag{127}$$

As such, the wavefunctional provides a representation for an arbitrary quantum mechanical state $|\Psi(\eta)\rangle$.

This sets a functional Schrödinger formulation for studying interactions where the time evolution of the state $|\Psi(t)\rangle$ is governed by the Schrödinger equation. The measure $\mathcal{D}\phi(\vec{x}, \eta)$ takes the reality conditions of the field into account. In working with Fourier modes $\phi_{\vec{k}}$, defined by

$$\phi(\vec{x}, t) = \int \frac{d^3k}{(2\pi)^3} e^{i\vec{k}\cdot\vec{x}} \phi_{\vec{k}}(\eta) \tag{128}$$

the reality of the field then implies $(\phi_{\vec{k}})^* = \phi_{-\vec{k}}$. In this case, the measure is to include both $\phi_{\vec{k}}$ and $\phi_{\vec{k}}^*$ as well as the reality condition. We will make use of this in our discussion below.

Since the late-time boundary of de Sitter is a convenient place to recognize the unitary irreducible representations of the de Sitter group, we will focus on the late-time behaviour of the wavefunction. We will consider the Bunch–Davies wavefunction which is a functional of some given late-time profile which we will denote by $\Phi$ and the late-time $\eta_0$ itself, $\Psi_{BD}[\Phi, \eta_0]$. We can also talk about the Fourier modes $\Phi_{\vec{k}}$, of the late-time profile which we will make use of below.

This specification of the wavefunction is partly to do with the boundary conditions. The Bunch–Davies wavefunction is obtained as a pathintegral over all field configurations that satisfy Bunch–Davies boundary conditions [59]

$$\Psi[\Phi, \eta_0] = \int_{\mathcal{C}} \mathcal{D}\phi e^{iS[\phi]}. \tag{129}$$

These boundary conditions require the field to be well behaved at early times and to approach the given profile $\Phi$ at late-times. The early-time limit $\eta \to -\infty$ requires the inclusion of a factor of $\pm i\epsilon$ to be well defined. The choice of the sign here defines two different contours

$$\mathcal{C}_- : \eta \in (-(1 + i\epsilon)\infty, \eta_0], \tag{130}$$

$$\mathcal{C}_+ : \eta \in (-(1 - i\epsilon)\infty, \eta_0]. \tag{131}$$

The mode solutions that satisfy these boundary conditions involve Hankel functions and the choice of the contour determines whether to pick Hankel functions of the first or second kind. We will work with the contour $\mathcal{C}_-$ only and therefore use Hankel functions of the first kind. Reference [60] works with the other contour as well.

In practice, it is generally difficult to perform the full integral in (130) and one consults to a semiclassical approximation in which the onshell action $S_{onshell}$ gives the dominant contribution to the path integral [37]. With our choice of the contour

$$\Psi[\Phi, \eta_0] = \int_{\substack{\phi(\eta) \to 0 \\ \text{as } \eta \to -(1+i\epsilon)\infty}}^{\phi(\eta_0) = \Phi} \mathcal{D}\phi \, e^{iS[\phi]} \sim \mathcal{N}_c(\eta_0) e^{iS_{onshell}[\Phi, \eta_0]}, \tag{132}$$

where $\mathcal{N}_c(\eta_0)$ denotes a time dependent normalization, the onshell action in $k-$space is given by

$$S_{onshell} = -\frac{1}{2} \int \frac{d^3k}{(2\pi)^3} \int_{\mathcal{C}_-} d\eta \frac{\partial}{\partial \eta} \left[ a(\eta)^2 \phi'_{\vec{k}}(\eta) \phi_{-\vec{k}}(\eta) \right], \tag{133}$$

with $a(\eta_0) = \frac{1}{H|\eta_0|}$. The late-time profile can also be decomposed in terms of Fourier modes $\Phi_{\vec{k}}$ and the solution that satisfies the boundary conditions can be written as

$$\phi_{\vec{k}} = \Phi_{\vec{k}} \frac{v_k(\eta)}{v_k(\eta_0)} \tag{134}$$

with $v_k(\eta) = |\eta|^{\frac{3}{2}} \mathcal{H}(k|\eta|)$ where $\mathcal{H}(k|\eta|)$ stands for the appropriate Hankel function depending on the mass, that is $H_\nu^{(1)}(k|\eta|)$ for light fields and $\tilde{H}_\rho^{(1)}(k|\eta|)$ for heavy fields. The ratio $\frac{v_k(\eta)}{v_k(\eta_0)}$ is called the *bulk-to-boundary propagator*. With all these considerations, the wavefunction in the late-time limit for a free field is a Gaussian of the late-time profile $\Phi_{\vec{k}}$, and accommodates $k$ and $\eta_0$ dependence as determined by the mode functions

$$\Psi_{BD}[\Phi, \eta_0] = \mathcal{N}_c(\eta_0) e^{-\frac{1}{2} \int \frac{d^3k}{(2\pi)^3} \mathcal{P}(k, \eta_0) \Phi_{\vec{k}} \Phi_{-\vec{k}}}. \tag{135}$$

The specific momentum and time dependence of $\mathcal{P}(k, \eta_0)$ differs for principal and complementary series representations. The momentum and time dependence of $\mathcal{P}$ determines the momentum and time dependence of the late-time correlation functions.

Paying attention to the behaviour of the Hankel functions in the late-time limit the principal and complementary series wavefunctions at late-times have the following forms [3]

$$\Psi_{BD}^{princip} = \mathcal{N}_p(\eta_0) \exp \left\{ \frac{|\eta_0|^{-3}}{2H^2} \int \frac{d^3k}{(2\pi)^3} \left[ i\frac{3}{2} + \rho \frac{1 + e^{-\rho\pi - 2i\gamma_\rho} \left( \frac{k|\eta_0|}{2} \right)^{2i\rho}}{1 - e^{-\rho\pi - 2i\gamma_\rho} \left( \frac{k|\eta_0|}{2} \right)^{2i\rho}} \right] |\Phi_{\vec{k}}|^2 \right\}, \tag{136}$$

$$\Psi_{BD}^{compl}[\Phi; \eta_0] = \mathcal{N}_c(\eta_0) \exp \left\{ \frac{|\eta_0|^{-3}}{2H^2} \int \frac{d^3k}{(2\pi)^3} \left[ i\frac{3}{2} + i\nu \frac{\frac{1 + i\cot(\pi\nu)}{\Gamma(\nu+1)} \left( \frac{k|\eta_0|}{2} \right)^{2\nu} + i\frac{\Gamma(\nu)}{\pi}}{\frac{1 + i\cot(\pi\nu)}{\Gamma(\nu+1)} \left( \frac{k|\eta_0|}{2} \right)^{2\nu} - \frac{i\Gamma(\nu)}{\pi}} \right] |\Phi_{\vec{k}}|^2 \right\}. \tag{137}$$

The late-time correlation functions can be obtained from the wavefunction by

$$\langle \mathcal{O}_{\vec{k}_1}^{(1)} \dots \mathcal{O}_{\vec{k}_n}^{(n)} \rangle = \frac{1}{\mathcal{N}(\eta_0)} \int_{\substack{\text{configurations} \\ \text{where } \Phi_{\vec{k}}^* = \Phi_{-\vec{k}}}} [\mathcal{D}\Phi_{\vec{k}}] \Psi_{BD}^*[\Phi_{\vec{k}}] \mathcal{O}_{\vec{k}_1}^{(1)} \dots \mathcal{O}_{\vec{k}_n}^{(n)} \Psi_{BD}[\Phi_{\vec{k}}]. \quad (138)$$

For the operators $\mathcal{O}_{\vec{k}}$, we will be interested in correlators of both field $\Phi_{\vec{k}}$ and its canonical conjugate momenta $\Pi_{\vec{k}}$,

$$\mathcal{O}_{\vec{k}} = \{\Phi_{\vec{k}}, \Pi_{\vec{k}}\}. \quad (139)$$

In momentum space the canonical conjugate momentum operator is represented by [61]

$$\Pi_{\vec{k}} = \frac{1}{i} (2\pi)^3 \frac{\delta}{\delta \Phi_{-\vec{k}}}. \quad (140)$$

These two operators have the following nontrivial commutation relation

$$[\Phi_{\vec{k}}, \Pi_{\vec{k}'}] = i(2\pi)^3 \delta^{(3)}(\vec{k} + \vec{k}'). \quad (141)$$

With this formalism, one can obtain the momentum dependence of the late-time correlators explicitly. However, in the wavefunction picture it is not clear how the late-time operators contribute. The canonical quantization calculations help to establish the relation with the unitary irreducible representations [3].

In the initial discussion of wavefunction methods for de Sitter Holography and the computation of inflationary correlators as an application, in [49] the form of the wavefunction in terms of the boundary operators were given as part of the dictionary. In this discussion the contribution of the boundary operators to the wavefunction is fixed based on intuition from AdS/CFT Holography. In otherwords it is based on intuition from the case of a negative cosmological constant. Recent investigations of [3,60] show that operators that live at the late-time boundary of de Sitter and that can be recognized as unitary irreducible representations of the de Sitter group contribute in a different form to the wavefunction then was considered in [49]. Initially reference [60] pointed out that there is a significant difference when discussing principal series fields while the complementary series fields can still be captured by a form of the wavefunction that is similar to what arises in AdS/CFT. Looking at (136) and (137) the two wavefunctions have a similar structure of momentum dependence, where the terms in the numerator and denominator capture contributions from both of the normalized late-time opertors $\alpha_N$ and $\beta_N$. The resemblance between the complementary series wavefunction and the AdS/CFT partition function arises when in the late-time limit one starts neglecting contributions with a positive exponent of $\eta_0$, that come with $|\eta_0|^\nu$ next to contributions with $|\eta_0|^{-\nu}$. This is possible only in the complementary series case because the exponent there is real. Reference [60] carries on this simplification at the level of the wavefunction while reference [3] keeps these terms so as to set a better comparison.

As we said, the contribution of the boundary operators is not explicit in the wavefunction itself. The wavefunction makes the $k-$dependence explicit. Hints about the contribution of the boundary operators are better established through the two-point functions. This connection becomes possible by noting the relation between the two-point functions computed in canonical quantization and wavefunction methods individually since canonical quantization can keep explicit track of the boundary operators as demonstrated for the late-time operators in [3]. With this insight the late-time operators contribute to the two-point function and hence the wavefunction as follows. In the case of principal series representations we obtain

$$\langle \Phi^H_{\vec{k}} \Phi^H_{\vec{k}'} \rangle = \frac{(2\pi|\eta_0|)^3 H^2}{4\rho \sinh(\rho\pi)} \delta^{(3)}(\vec{k} + \vec{k}') \times$$

$$\left[ 2\cosh(\rho\pi) - e^{2i\gamma_\rho} \left( \frac{k|\eta_0|}{2} \right)^{-2i\rho} - e^{-2i\gamma_\rho} \left( \frac{k|\eta_0|}{2} \right)^{2i\rho} \right] \quad (142)$$

$$\langle \Pi^H_{\vec{k}} \Pi^H_{\vec{k}'} \rangle = \frac{(2\pi)^3 \delta^{(3)}(\vec{k} + \vec{k}')}{|\eta_0|^3 H^2} \frac{1}{4\rho \sinh(\rho\pi)} \times \left[ \left( \frac{3}{2} - i\rho \right) \left( \frac{3}{2} + i\rho \right) 2\cosh(\rho\pi) \right.$$

$$\left. - \left( \frac{3}{2} - i\rho \right)^2 e^{2i\gamma_\rho} \left( \frac{k|\eta_0|}{2} \right)^{-2i\rho} - \left( \frac{3}{2} + i\rho \right)^2 e^{-2i\gamma_\rho} \left( \frac{k|\eta_0|}{2} \right)^{2i\rho} \right], \quad (143)$$

where $\gamma_\rho$ is defined via [62]

$$\Gamma(1 + i\rho) = \sqrt{\frac{\pi\rho}{\sinh(\pi\rho)}} e^{i\gamma_\rho}. \quad (144)$$

In the case of complementary series representations, we have

$$\langle \Phi^L_{\vec{k}} \Phi^L_{\vec{k}'} \rangle = \frac{\pi}{4} (2\pi|\eta_0|)^3 H^2 \delta^{(3)}(\vec{k} + \vec{k}') \times$$

$$\times \left[ \frac{1 + \cot^2(\nu\pi)}{\Gamma^2(\nu+1)} \left( \frac{k|\eta_0|}{2} \right)^{2\nu} + \frac{\Gamma^2(\nu)}{\pi^2} \left( \frac{k|\eta_0|}{2} \right)^{-2\nu} - \frac{2\cot(\nu\pi)}{\nu\pi} \right] \quad (145)$$

$$\langle \Pi^L_{\vec{k}} \Pi^L_{\vec{k}'} \rangle = \frac{\pi}{4} \frac{(2\pi)^3 \delta^{(3)}(\vec{k} + \vec{k}')}{|\eta_0|^3 H^2} \times \left[ -\frac{2\cot(\nu\pi)}{\nu\pi} \left( \frac{3}{2} + \nu \right) \left( \frac{3}{2} - \nu \right) \right.$$

$$\left. + \frac{1 + \cot^2(\nu\pi)}{\Gamma^2(\nu+1)} \left( \frac{3}{2} + \nu \right)^2 \left( \frac{k|\eta_0|}{2} \right)^{2\nu} + \frac{\Gamma^2(\nu)}{\pi^2} \left( \frac{3}{2} - \nu \right)^2 \left( \frac{k|\eta_0|}{2} \right)^{-2\nu} \right]. \quad (146)$$

What we call the complementary series wavefunction also captures the massless scalar wavefunction which we now know to belong to the discrete series representations. In this case

$$\langle \Phi^L_{\vec{k}} \Phi^L_{\vec{k}'} \rangle = \frac{\pi}{4} (2\pi|\eta_0|)^3 H^2 \delta^{(3)}(\vec{k} + \vec{k}') \left[ \frac{1}{\Gamma^2(5/2)} \left( \frac{k|\eta_0|}{2} \right)^3 + \frac{\Gamma^2(3/2)}{\pi^2} \left( \frac{k|\eta_0|}{2} \right)^{-3} \right] \quad (147a)$$

$$\langle \Pi^L_{\vec{k}} \Pi^L_{\vec{k}'} \rangle = (2\pi)^3 \delta^{(3)} \left( \vec{k} + \vec{k}' \right) \frac{k^3}{2H^2}. \quad (147b)$$

The two-point functions above match our results for the canonically quantized mode functions. Emphasizing the massless case separately within our labeling convention from Section 4.1 with

$$p = \{L, H, M\}, \quad \mu_p = \begin{cases} \mu_L = \nu \\ \mu_H = i\rho \\ \mu_M = \frac{3}{2} \end{cases} \quad (148)$$

everything can be written collectively as

$$\langle \Phi_{\vec{k}} \Phi_{\vec{k}'} \rangle = \langle \varphi^{lt}_{\vec{k}} \varphi^{lt}_{\vec{k}'} \rangle \quad (149)$$

$$= H^2 |\eta_0|^3 \left\{ c^p_1 |\eta_0|^{-2\mu_p} \langle \alpha_N \alpha_N \rangle + c^p_2 |\eta_0|^{2\mu_p} \langle \beta_N \beta_N \rangle + c^p_3 [\langle \alpha_N \beta_N \rangle + \langle \beta_N \alpha_N \rangle] \right\}$$

$$\langle \Pi_{\vec{k}} \Pi_{\vec{k}'} \rangle = \langle \pi^{lt}_{\vec{k}} \pi^{lt}_{\vec{k}'} \rangle$$

$$= \frac{1}{|\eta_0|^3 H^2} \left\{ c_1^p |\eta_0|^{-2\mu_p} \left( \frac{3}{2} - \mu_p \right)^2 \langle \alpha_N \alpha_N \rangle + c_2^p |\eta_0|^{2\mu_p} \left( \frac{3}{2} + \mu_p \right)^2 \langle \beta_N \beta_N \rangle \quad (150)$$

$$+ c_3^p \left( \frac{3}{2} - \mu_p \right) \left( \frac{3}{2} + \mu_p \right) [\langle \alpha_N \beta_N \rangle + \langle \beta_N \alpha_N \rangle] \right\}$$

where we have suppressed the label $p$ for $\Phi_{\vec{k}}$, $\Pi_{\vec{k}}$, $\varphi^{lt}_{\vec{k}}$, $\pi^{lt}_{\vec{k}}$, $\alpha_N$, $\beta_N$ in the interest of space and $c_i^p$ implies that the coefficients differ depending on the category. In the massless scalar case, the conjugate momentum two-point function depends only on the late-time operator $\beta_N^M$, and this is not due to any simplifications. At the late-time boundary we find two noncommuting operators with different dimensions $\{\alpha_N, \beta_N\}$. Canonically we also have two noncommuting operators the field and its conjugate momenta. Additionally, although we are interested in the late-time limit, we are aware of the bulk behaviour of field and conjugate momentum operators as well. Addressing the conjugate momentum correlators as well as the field correlators helps establish the full picture and interpret both of the late-time operators better.

The simplification that makes the complementary series wavefunction look similar to the AdS/CFT partition function implies neglecting the contributions coming from the $\beta_N^L$ operators with the higher scaling dimension $\Delta = \frac{3}{2} + \nu$ next to the operator with the lower scaling dimension $\alpha_N^L$ with $\Delta = \frac{3}{2} - \nu$.

Wavefunction techniques provide one of the valuable tools towards de Sitter holography. So let us end this section with a small review on de Sitter Holography. Earlier hints towards approaching de Sitter physics within holography come from calculations of entropy for asymptotically $dS_3$ spacetimes and identifying central charges in the asymptotic algebras involved [63–65]. A key point in this line of research is the equivalence between Einstein gravity in three dimensions with a positive cosmological constant and the Chern Simons theory. This line of research hints that the idea of Holography, initially introduced through a specific construction that involves a gravitational theory on Anti de Sitter spacetime, which corresponds to negative cosmological constant, and a conformal field theory (CFT) [66], might be a tool applicable within a more general context. An early work along this line for the case of the positive cosmological constant is the proposal of $Chern - Simons_{3+1}/CFT$ correspondence [67]. Today holography is indeed a tool whose general properties for all values of the cosmological constant are being explored from various angles. Further hints towards holography in the case of the positive cosmological constant came from studies at the early-time boundary of de Sitter [68], on the construction of a quantum Hilbert space [69], and on correlation functions that can have observable signatures in terms of inflationary correlation functions [49]. Each of these directions emphasize a different feature: features of the boundary, features of the Hilbert space and features of the wavefunction. Today the hints from the equivalence with Chern-Simons theory can be used to compute specific contributions to the de Sitter horizon entropy in three dimensions due to massless scalar fields in static patch [70]. The hints on the features of the Hilbert space have reached a stage of both providing insight on possible realizations of the dS/CFT proposal [34] as well as on having a BRST interpretation of de Sitter observables [71]. This later BRST interpretion can also help in addressing how to handle dynamical gravity within de Sitter Holography. The hints on correlation functions are developing into a means of interpreting why inflationary spectra may be the way it is observed [51] and how it can be further dissected in terms of what degrees of freedom can contribute to it [40].

Lastly, one may want to compare the recent advances on holography in the case of a negative cosmological constant with its positive cosmological constant counterpart. In the case of a negative cosmological constant, recent attention have been pointed towards new saddles to the gravitational path integral via the method of replica wormholes, that bring

new insight on issues related with unitarity in black hole thermodynamics [72]. The closest equivalent to this development that we are aware of from the positive cosmological side is the work of [73], where contributions of compact manifolds to the Euclidean gravitational path integral in the case of a positive cosmological constant are considered and, in addition, a Lorentzian perspective is studied. A second example is reference [74], that addresses horizon entropy in the case of a positive cosmological constant. In this review, we have focused on correlation functions; effects of these recent developments would be seen in higher order correlation functions rather then the two-point functions we have considered.

## 6. Discussion

The late-time operators we used for the purpose of highlighting the features of unitary irreducible representations of the de Sitter group, which in Wigner's classification of particles would be the particles of a de Sitter universe, provide a contribution to the recent developments in quantum field theory and holography in the case of a positive cosmological constant. Even the consideration of scalar fields alone, which are the observable degrees of freedom that are easiest to access in cosmological surveys, have enough variety to give an example to each one of the categories except for exceptional series. This is a contribution from the late-time boundary. To place what we have already said within the bigger picture, let us summarize some of the other recent contributions to the discussion.

Discussions focusing on the late-time boundary include the structure of the Bunch–Davies wavefunction [48], and its organization for more general FLRW spacetimes [75], as well as introduction of weight-shifting operators towards implementing bootstrap methods that are well developed mainly for conformal field theories other then those that arise for a de Sitter setting [76]. Other advances along the application of non-perturbative bootstrap methods in the case of a positive cosmological constant involve [77] as well as the perturbative treatment of [78] both of which emphasize the role of the principal series unitary irreducible representations of the de Sitter group, with focus on Källén–Lehmann decomposition for correlators in the case of two dimensions. Considering interactions from the point of view of tensor product states, ref. [30] starts a discussion of the tensor product of the late-time operators we discussed in this review, by considering the case of principal series representations, following [11].

Principal series representations have also made an appearance in holographic constructions with a positive cosmological constant. Some examples on this front are [79], which considers the case of three dimensions with remarks on the cluster decomposition properties of CFTs that would host these kind of operators where the bulk operators are obtained from operators at the early-time boundary. Another example is the correspondence of principal series representations in two dimensional de Sitter with a specific quantum mechanical model given in [29]. In two dimensions, studies of flow geometries [80,81], and matrix models [82] have also brought new insight on aspects of de Sitter holography and its features that set it apart from holography with a negative cosmological constant.

The shadow late-time operators we came across in our discussion of complementary series representations are representations that are equivalent to their nonshadow counter parts. This equivalence is understood by comparing the trace of the two representations. The trace is called the *character* of the representation. The intertwining maps we discussed set a similarity transformation, leaving the trace invariant. We have not provided any explicit character calculations in this review; however, significant progress has been made in this front recently, in the static patch of de Sitter, where the characters of the representations have been shown to encode information on the static patch horizon entropy [74]. It would be an interesting objective to understand to what physical quantity at the late-time boundary the characters would contribute to.

Other recent developments in the static patch which we have not yet mentioned include [83] on Markovian dynamics through considerations of light scalars, ref. [84] on construction of higher spin quasi normal modes and their categorization in terms of de Sitter representations, ref. [85] on scattering of a conformally massless scalar with a cubic

interaction, ref. [86] on the consideration of Källén–Lehmann decomposition. As this is the patch relevant for a physical observer living in de Sitter, further analysis from the group theory frontier on this patch will let us understand better the particles at the disposal of an observer living in the de Sitter universe, such as ourselves in today's Dark Energy dominated epoch of cosmology.

A full list of particles is nowhere complete without fermions and gauge fields. The aim of this review was to point towards the categories of unitary irreducible representations of the de Sitter group and since the late-time behaviour of free scalars provided an example to each category we have framed our discussion around scalars. Reference [87] is the earliest work we are aware of that adresses fermions by considering spin$-\frac{1}{2}$ wave equation on de Sitter. Since then, gauge fields and fermions of other spin have also been brought into the discussion. This discussion has reached a point where today their Wightman propagators can conveniently be considered within embedding space formalism [28]. One can also check [28] for a further list of studies of spinors on de Sitter spacetime.

In our discussion of late-time operators, we paid particular attention to their normalization. Normalization of states that can arise from fields on de Sitter have also been a guiding property in obtaining ranges of masses for gauge fields [88] or dimensionality for fermions [35]. Reference [88] studies allowed mass ranges for scalar, vector and spin two fields on de Sitter by considering equal-time commutations in Poincaré patch and analysing the norm of the states these fields give rise to. As a result, for the mass $M$ in terms of the value of the cosmological constant $\Lambda$, the range $0 < M^2 < \frac{2}{3}\Lambda$ is forbidden for spin$-2$ as it gives rise to negative norm states. Reference [35] analysis covers spin$-\frac{3}{2}$ and spin$-\frac{5}{2}$ in terms of the unitary irreducible representations of $Spin(d+1,1)$, and the double cover of de Sitter group in general dimensions using analytic continuation methods from the sphere to de Sitter. To be more precise, it is the strictly massless spin$-\frac{3}{2}$, strictly and partially massless spin$-\frac{5}{2}$ fields that are studied. A key point in the discussion also relies on the normalizability properties of these fields. The result is that only in four spacetime dimensions one can have these degrees of freedom be unitary. In other dimensions they give rise to negative norm states. The analysis also suggests that these spin$-\frac{3}{2}$ and spin$-\frac{5}{2}$ fields belong to the discrete series representations. On one hand, the categorization of unitary irreducible representations of de Sitter recognize these fields as fermionic gauge fields. On the other hand, the massless spin$-\frac{3}{2}$ field is the gravitino field, known to be a fermionic gauge field in supersymmetry literature. From this perspective the results of [35] supported by representation theory of the de Sitter group, connect supersymmetry with four spacetime dimensions in the presence of the cosmological constant. We leave it to future studies to extend our list of late-time operators to include cases of nonzero spin. The efforts towards enlarging the list of late-time operators with this perspective involves understanding how to address dynamical gravity as well.

All in all, it seems there is a lot of new insight we can gain on the presence of a positive cosmological constant by exploring further the merits of the unitary irreducible representations it can accommodate.

**Funding:** This research is built on some of the results obtained during project SymAcc funded by the European Union's Horizon 2020 research and innovation programme under the Marie Skłodowska-Curie grant agreement No 840709 and it is funded in the first part by the European Structural and Investment Funds and the Czech Ministry of Education, Youth and Sports (MSMT) under Project CoGraDS with grant number-CZ.02.1.01/0.0/0.0/15003/0000437) and in the second part by TÜBİTAK (the Scientific and Technological Research Council of Turkey) 2232—B International Fellowship for Early Stage Researchers programme with project number 121C138.

**Data Availability Statement:** Not applicable.

**Acknowledgments:** The author sincerely thanks Constantinos Skordis, Dionysios Anninos, Tarek Anous, Benjamin J. Pethybridge, Vasileios A. Letsios, Taha Ayfer for insightful discussions during the course of this work.

**Conflicts of Interest:** The author declares no conflict of interest. The funders had no role in the design of the study; in the writing of the manuscript; or in the decision to publish the results.

## Abbreviations

The following abbreviations are used in this manuscript:

| | |
|---|---|
| dS | de Sitter spacetime |
| AdS | Anti de Sitter spacetime |
| FLRW | Friedmann–Lemaître–Robertson–Walker metric |
| CFT | Conformal Field Theory |

## Appendix A. Killing Vectors of de Sitter in Different Coordinates

*Appendix A.1. The Absence of a Global Time-Like Killing Vector*

The Killing equations for de Sitter in global coordinates are

$$\partial_T \xi_T = 0 \tag{A1a}$$

$$\partial_{\theta_1} \xi_T + \partial_T \xi_{\theta_1} - 2H tanh(HT)\xi_{\theta_1} = 0 \tag{A1b}$$

$$\partial_{\theta_2} \xi_T + \partial_T \xi_{\theta_2} - 2H tanh(HT)\xi_{\theta_2} = 0 \tag{A1c}$$

$$\partial_{\theta_3} \xi_T + \partial_T \xi_{\theta_3} - 2H tanh(HT)\xi_{\theta_3} = 0 \tag{A1d}$$

$$\partial_{\theta_1} \xi_{\theta_1} - \frac{1}{H} cosh(HT) sinh(HT)\xi_T = 0 \tag{A1e}$$

$$\partial_{\theta_2} \xi_{\theta_1} + \partial_{\theta_1} \xi_{\theta_2} - 2cot\theta_1 \xi_{\theta_2} = 0 \tag{A1f}$$

$$\partial_{\theta_3} \xi_{\theta_1} + \partial_{\theta_1} \xi_{\theta_3} - 2cot\theta_1 \xi_{\theta_3} = 0 \tag{A1g}$$

$$2\partial_{\theta_2} \xi_{\theta_2} + sin2\theta_1 \xi_{\theta_1} - \frac{1}{H} sin^2\theta_1 sinh(2HT)\xi_T = 0 \tag{A1h}$$

$$\partial_{\theta_3} \xi_{\theta_2} + \partial_{\theta_2} \xi_{\theta_3} - 2cot\theta_2 \xi_{\theta_3} = 0 \tag{A1i}$$

$$\partial_{\theta_3} \xi_{\theta_3} + sin\theta_2 \left( cos\theta_2 \xi_{\theta_2} + \frac{1}{H} sin\theta_1 sin\theta_2 \left[ -\frac{1}{2} sin\theta_1 sinh(2HT)\xi_T + H cos\theta_1 \xi_{\theta_1} \right] \right) = 0 \tag{A1j}$$

These equations determine the components $\xi_\mu$ of the Killing vector. A time-like Killing vector, that would generate time translations will be of the form $\xi^T \partial_T$, with components $\xi^\mu = \{\xi^T = constant, 0, 0, 0\}$. Since the de Sitter metric in global coordinates is diagonal, $\xi^T = g^{TT}\xi_T$ and we can just check if $\xi_\mu = \{1, 0, 0, 0\}$ is a solution to (A1a)–(A1j). From Equations (A1e), (A1h) and (A1j) we immediately see that this is not one of the solutions. In addition, using the metric compatibility one can rewrite the Killing equations for components $\xi^\mu$. Either way, one cannot find a time-like Killing vector for de Sitter in global coordinates.

On the other hand, the dilatation and special conformal transformation Killing vectors which we listed in Section 2 in Poincaré patch have timelike components which carry time and spatial dependence. We will explore these Killing vectors further in the next section in the embedding space formalism where $X^A$ are embedding space coordinates, and for instance $\chi^0(X^A)$ for these. However, the dilatation and special conformal transformation Killing vectors also have nontrivial components along spatial directions. In Appendix A.2, we write explicit expressions in the Poincaré patch coordinates but one can find the expressions of embedding space coordinates in terms of global patch coordinates in the references we give.

*Appendix A.2. Killing Vectors in Poincaré Patch and a Review of Embedding Space Formalism*

Constant curvature spacetimes can be embedded into a flat spacetime of one higher dimension. This applies both to dS and AdS. This is why the isometries of these spacetimes are Lorentz groups that we are familiar with from flat spacetime quantum field theory. However, whether the higher dimension is spacelike or timelike depends on the sign of the curvature. For positive constant curvature, the embedding is achieved by adding

a spacelike dimension. This is why the symmetry group of $d + 1$ dimensional de Sitter is $SO(d + 1, 1)$. In the case of negative constant curvature the embedding requires an extra timelike dimension. In this case, the isometry group is $SO(d, 2)$. This approach is called the *embedding space* or *ambient space* approach. It is an approach that offers technical ease especially when considering a coordinate free formalism or general spin. Being coordinates of a flat spacetime, the embedding space coordinates are raised and lowered by the flatspace metric $\eta_{AB} = diag(-1, 1, 1, 1, 1)$. If we denote embedding space coordinates by $X^A$ and the spacetime coordinates by $x^\mu$, the two set of coordinates are related to each other $X^A(x^\mu)$ such that

$$X^A X_A = -(X^0)^2 + (X^i)^2 + (X^4)^2 = \frac{1}{H^2},$$
(A2)

for de Sitter, where $i = 1, 2, 3$. Some reviews on the embedding space formalism for de Sitter in general dimensions are [28,77,78,89]. Here, we will follow the conventions of [28]. The embedding space formalism for AdS is reviewed in [90,91].

For each coordinate patch, the relation $X^A(x^\mu)$ differs. For the Poincaré Patch, the coordinates of the embedding space and the de Sitter coordinates are related as

$$X^0 = \frac{1 + x^2 - \eta^2}{2H\eta}, \ X^i = \frac{x^i}{H\eta}, \ X^4 = \frac{1 - x^2 + \eta^2}{2H\eta}$$
(A3a)

$$\eta = \frac{1}{H(X^4 + X^0)}, \ x^i = \frac{X^i}{X^4 + X^0}$$
(A3b)

where $x^2 = \eta_{ij} x^i x^j$. If we denote generators in embedding space by $L_{AB}$ and generators on de Sitter slice by $\mathcal{L}_{\mu\nu}$, the antiHermitian generators in the embedding space are defined as

$$L_{AB} = (X_A \partial_B - X_B \partial_A).$$
(A4)

Objects on the de Sitter slice with coordinates $x^\mu, \mu = 0, 1, 2, 3$ and objects on the higher dimensional embedding space with coordinates $X^A, A = 0, 1, 2, 3, 4$ are related to each other via

$$e_\mu{}^A \equiv \frac{\partial X^A}{\partial x^\mu},$$
(A5)

which pushes forward the embedding space fields to the de Sitter slice. Note that not all embedding space fields need to correspond to physical fields on the de Sitter slice. Necessary conditions such as transversality need to be taken into account when constructing de Sitter fields from ambient space fields. For our purposes, the push-forward is enough for us to access the isometry generators. The generators on the de Sitter slice $\mathcal{L}$ can be obtained from the generators on the embedding space $L_{AB}$ as follows

$$\mathcal{L}_{\mu\nu} = e_\mu{}^A e_\nu{}^B L_{AB}.$$
(A6)

Let us do this explicitly to demonstrate the link between $L_{04}$ and the dilatation Killing vector. In terms of embedding space coordinates we have

$$L_{04} = (X_0 \partial_4 - X_4 \partial_0).$$
(A7)

The partial derivatives in embedding space coordinates are evaluated as follows in terms of de Sitter coordinates

$$\frac{\partial}{\partial X^0} = \frac{\partial x^\mu}{\partial X^0}\frac{\partial}{\partial x^\mu} = -H\eta^2\partial_\eta - H\eta x^j\partial_j \tag{A8a}$$

$$\frac{\partial}{\partial X^4} = \frac{\partial x^\mu}{\partial X^4}\frac{\partial}{\partial x^\mu} = -H\eta^2\partial_\eta - H\eta x^j\partial_j \tag{A8b}$$

$$\frac{\partial}{\partial X^i} = \frac{\partial x^\mu}{\partial X^i}\frac{\partial}{\partial x^\mu} = H\eta\partial_i \tag{A8c}$$

Since the de Sitter coordinates $\{\eta, x^i\}$ have a similar dependence on the embedding coordinates $\{X^0, X^4\}$ the expressions above equal one another. Now, using (A3a), (A3b) and (A8) we can rewrite (A7) in terms of coordinates on the de Sitter slice as

$$D \equiv L_{04} = \left(\eta\partial_\eta + x^i\partial_i\right) = \xi^{(D)} \tag{A9}$$

which is the Dilatation Killing vector in Poincaré coordinates. Note that $L_{40} = -\xi^{(D)}$, which is the way we introduced the relation between the generators $\mathcal{L}^{(B)}$ and Killing vectors $\xi^{(B)}$ in Section 2.

Following through similar arguments, the special conformal transformations correspond to

$$C_i \equiv L_{0i} + L_{4i} = \left((\eta^2 - x^2)\partial_i + 2x_i(\eta\partial_\eta + x^j\partial_j)\right) = -\xi^{(SCT)}, \tag{A10}$$

translations correspond to

$$P_i \equiv L_{0i} - L_{4i} = -\partial_i = -\xi^{(tr)} \tag{A11}$$

and rotations correspond to

$$M_{ij} = L_{ij} = i\left(x_i\partial_j - x_j\partial_i\right) = i\xi^{(rot-\hat{i}\times\hat{j})}. \tag{A12}$$

The quadratic Casimir in terms of the antiHermitian generators is

$$\mathcal{C} = -\frac{1}{2}L_{AB}L^{AB} \tag{A13}$$

$$= \frac{1}{2}\left[C_xP_x + P_xC_x + C_yP_y + P_yC_y + C_zP_z + P_zC_z\right] + D^2 - \frac{1}{2}M_{ij}^2 \tag{A14}$$

$$= C_iP_i + 3D + D^2 - \frac{1}{2}M_{ij}^2 \tag{A15}$$

where we made use of the commutation that $[C_i, P_i] = 2D$. Then the Casimir eigenvalue for $SO(4,1)$ is [11]

$$\mathcal{C} = l(l+1) + c^2 - \frac{9}{4}. \tag{A16}$$

In addition to the subgroups we listed, there is also the maximally compact subgroup $SO(d+1)$ in general dimensions, $SO(4)$ for four spacetime dimensions. The generators of $SO(4)$ are $L_{AB}$ with $A, B = 1, 2, 3, 4$. Notice that the maximally compact subgroup is a collection of the rotation subgroup generators and linear combinations of translation and special conformal transformation generators. Among the maximally compact subgroup generators, the generators $L_{12}, L_{13}, L_{23}$, are also the $SO(3)$ generators we denoted by $M_{ij}$ in Equation (A12). From Equations (A10) and (A11), we see that the generators $L_{41}, L_{42}, L_{43}$, are the following linear combination of translation and special conformal transformation generators

$$L_{4i} = \frac{C_i - P_i}{2}. \tag{A17}$$

Let us end this appendix by briefly mentioning that in considering physical fields in terms of embedding space fields, there are two ways to make the connection. Written as

tensors, the physical fields, which are symmetric traceless tensors, correspond to symmetric traceless tensors in embedding space that are transverse to the surface that defines the physical spacetime of interest. If we denote ambient space field by $\mathcal{A}$, the transversality condition is expressed as

$$X \cdot \mathcal{A}(X) = 0. \tag{A18}$$

In addition, one can use an index free formalism, by encoding the ambient space tensors into homogeneous polynomials of some degree.

The link between embedding space objects and physical fields are established via projection operators. Physical tensors are retrieved from the embedding space tensors via contraction with the projection operator $G_{AB}$. This is a symmetric transverse tensor whose defining property is

$$X^A G_{AB} = G_{AB} X^B = 0. \tag{A19}$$

The projection operator is given by the following expression

$$G_{AB} = \eta_{AB} - H^2 X_A X_B. \tag{A20}$$

In the index free formalism, ambient space tensors are retrieved by acting with a derivative operator $K_A$ on the homogeneous polynomial for the further details of which we refer the reader to [28]. An important property of the projection operators $G_{AB}$ and $K_A$ is that they ensure the resulting objects are transverse to the de Sitter slice in the embedding space.

## Notes

[1] To be more precise, this map involves mirror images and $\tilde{\chi} = \{\tilde{l}, \tilde{c}\}$ where $\tilde{l}$ is the mirror image of $l$. For further details of this map we refer the reader to [2] and references within. Scalar representations automatically satisfy $\tilde{l} = l = 0$. This is not the only case where the mirror image of a representation is equivalent to itself. Denoting the rank by $\nu$, the mirror image of any irreducible representation $l$ of $SO(2\nu + 1)$ is also equivalent to itself (see [11] Section 2.A for further details of this). In the case of four spacetime dimensions, the rotation subgroup $SO(3)$ falls into this category. The maximally compact subgroup $SO(4)$ does not. The discrete series representations are induced by the maximally compact subgroup $SO(4)$ and correspond to gauge fields which have spin higher then spin zero. One needs to be more careful about the equivalence of representations in the case of higher spin. Based on this while the intertwining map relates two equivalent representations in the case of complementary series, the representation and its shadow are inequivalent in the case of discrete series [11]. The equivalence of the representations depends on their trace, two equivalent representations have the same trace. We leave it for future studies to check what happens to the late-time operators for the massless scalar under mirror image, to be fully certain that they correspond to discrete series.

[2] We take the early-time limit as $\eta \to -\infty(1 + i\epsilon)$ and accordingly pick out Hankel functions of first kind. Another limit to consider would be $\eta \to -\infty(1 - i\epsilon)$, in which case one would pick up Hankel functions of second kind.

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
