# Peer review of "Particles of a de Sitter Universe"

_universe, doi:10.3390/universe9020059_

Round 1
Reviewer 2 Report
This article is an in-depth review of the representation theory of the de Sitter group, whose unitary irreps are relevant for (inflationary) cosmology and de Sitter holography. This review is highly detailed and will be a useful and accessible overview of techniques that have been developed. Overall, I think the review is well-written and provides a useful addition to the literature.
Some minor comments:
Perhaps a shortcoming is that one motivation to study the de Sitter group is de Sitter holography. However, correlation functions (a prime object of interest) are not observables in quantum gravity. One might therefore wonder what role the described developments will play in holography (with dynamical gravity). Are there developments on this front?
Related to this, there are recent developments on quantum extremal islands and replica wormholes. Will such effects, which contribute to the path integral, also influence correlation functions? Perhaps a few sentences addressing these points would add to the completeness.
Finally, two typo’s?
- In line 222 the different subgroups are mentioned. However, below I don’t see a discussion of the “maximally compact subgroup”.
- I think eq (11) should not have an absolute value. Otherwise the scaling weight can never be imaginary.
After the author addresses these points, I would be happy to recommend this paper for publication.
